# Impact of pulmonary African trypanosomes on the immunology and function of the lung

Dorien Mabille [1], Laura Dirkx [1], Sofie Thys[2], Marjorie Vermeersch[3,4], Daniel Montenye[3,4], Matthias Govaerts [1], Sarah Hendrickx[1], Peter Takac[5,6], Johan Van Weyenbergh[7], Isabel Pintelon[2], Peter Delputte [1], Louis Maes [1], David Pérez-Morga[3,4], Jean-Pierre Timmermans [2] & Guy Caljon [1] ✉

Approximately 20% of sleeping sickness patients exhibit respiratory complications, however, with a largely unknown role of the parasite. Here we show that tsetse fly-transmitted *Trypanosoma brucei* parasites rapidly and permanently colonize the lungs and occupy the extravascular spaces surrounding the blood vessels of the alveoli and bronchi. They are present as nests of multiplying parasites exhibiting close interactions with collagen and active secretion of extracellular vesicles. The local immune response shows a substantial increase of monocytes, macrophages, dendritic cells and γδ and activated αβ T cells and a later influx of neutrophils. Interestingly, parasite presence results in a significant reduction of B cells, eosinophils and natural killer cells. *T. brucei* infected mice show no infection-associated pulmonary dysfunction, mirroring the limited pulmonary clinical complications during sleeping sickness. However, the substantial reduction of the various immune cells may render individuals more susceptible to opportunistic infections, as evident by a co-infection experiment with respiratory syncytial virus. Collectively, these observations provide insights into a largely overlooked target organ, and may trigger new diagnostic and supportive therapeutic approaches for sleeping sickness.

T*rypanosoma brucei* is an obligate extracellular parasite responsible for human African trypanosomiasis (HAT) or sleeping sickness. The disease is restricted to sub-Saharan Africa where the tsetse fly vector (*Glossina* spp.) occurs. Upon the bite of an infected tsetse fly, infective metacyclic trypanosomes (MCF) are injected into the skin of the mammalian host[1,2]. From the skin microenvironment they enter the lymphatic system and colonize the bloodstream (stage-I). Upon further disease progression, parasites invade the central nervous system (stage-II) causing several severe neurological symptoms, including the typical disturbance of the circadian rhythm and death[1,3,4]. The therapies that are currently available to treat HAT are limited in number and face limitations of toxicity and drug resistance[5]. The implementation of fexinidazole, the first full oral treatment, and the prospect of a single-day oral treatment with acoziborole will likely completely change the nature of HAT treatment in the future[6,7], particularly in the absence of a protective vaccine despite decades of focused research. Major challenges towards elimination remain the still limited knowledge about the infection immunology and the existence of asymptomatic carriers sustaining transmission[8–10].

[1]Laboratory of Microbiology, Parasitology and Hygiene, Infla-Med Centre of Excellence, University of Antwerp, Wilrijk, Belgium. [2]Laboratory of Cell Biology and Histology, Department of Veterinary Sciences, University of Antwerp, Wilrijk, Belgium. [3]Center for Microscopy and Molecular Imaging, Université libre de Bruxelles, Gosselies, Belgium. [4]Laboratory of Molecular Parasitology, IBMM, Université libre de Bruxelles, Gosselies, Belgium. [5]Institute of Zoology, Slovak Academy of Sciences, 84506 Bratislava, Slovakia. [6]Scientica, Ltd., 83106 Bratislava, Slovakia. [7]Clinical and Epidemiological Virology, Department of Microbiology, Immunology and Transplantation, Rega Institute of Medical Research, KU Leuven, Leuven, Belgium. ✉e-mail: Guy.Caljon@uantwerpen.be

Recent work on tissue-resident trypanosomes has identified the dermis as a previously overlooked reservoir in asymptomatic individuals, which plays an important role in sustaining the transmission cycle[9,11]. Skin parasites were found to intricately interact with adipocytes supporting this particular tropism[11]. Tsetse-mediated infections of artificial human skin revealed that, after the establishment of a proliferative trypanosome population in the skin, the parasite appears to enter a quiescent state characterized by slow replication and reduced metabolism, which may reflect infection with low levels of inflammation and persistence in asymptomatic individuals[12]. Experimental mouse infections demonstrated that other tissues can also become heavily parasitized. The adipose tissue harbours high numbers of parasites that are metabolically distinct from the extracellular bloodstream forms (BSF). These metabolic adaptations to the local environment enable the use of host fatty acids as a carbon source which could participate in the weight loss associated with sleeping sickness[13].

During the haemolymphatic stage of the disease, patients show non-specific symptoms such as fever, headaches, joint pains and general malaise[1,14]. In addition to the characteristic neurological signs associated with stage-II HAT, patients occasionally exhibit organ-specific symptoms such as lymphadenopathy[15] and hepatosplenomegaly caused by tissue damage as a result of parasite-induced immune responses[16,17]. An underreported aspect is that a substantial proportion of HAT patients also suffer from respiratory symptoms. In Tanzania and Uganda more than 20% of stage-II patients were reported with a cough (20%) or dyspnoea (7%)[18,19]. Although the majority of severe dyspnoea cases were commonly attributed to cardiac insufficiency, the role of secondary viral/bacterial/fungal infections remains largely unknown. Bacterial bronchopneumonia is one of the reported complications in animal trypanosomiasis[20,21].

Our present work on *T. brucei* infection in mice following a natural infection by *Glossina morsitans* bites identified the lung tissue as a hitherto disregarded niche harbouring high parasite burdens during both early and established infections. We specifically focused on the importance of this tissue during infection and the associated immunological and functional changes.

## Results

### Tsetse-transmitted *T. brucei* parasites accumulate and expand in the lungs

Upon natural infection on the ear of mice by the bite of an infected *G. morsitans* tsetse fly, *T. brucei* burdens were determined using an SL-RNA RT-qPCR in different organs after prior vascular perfusion. The spleen appeared to be the main site of parasite accumulation during infection onset, but parasite levels are controlled upon disease progression. On the other hand, a steep increase in burden was observed in the sampled gonadal adipose tissue, as described previously[13]. Interestingly, lung-residing parasites constituted about a fifth of the total tissue burden both at infection onset and upon disease progression (Fig. 1a) without favouring a particular anatomical location (Fig. 1b). The relative importance of pulmonary trypanosome presence was further evidenced by the large increase in parasite burden over the course of infection (Fig. 1c) and a prominent BLI signal at the level of the thorax (Fig. 1d).

### Lung-resident trypanosomes are embedded in the tissue

Immunofluorescence staining of the variant-specific surface glycoprotein (VSG) coat of the parasite and the platelet endothelial cell adhesion molecule (PECAM-1 or CD31) on endothelial cells suggests extravascular localization of the parasite in near vicinity of the dense capillary network in the lung parenchyma (Supplementary Fig. S1). Fluorescence microscopy further confirmed an overall homogeneous distribution in the different lung sections. Scanning electron microscopy pinpointed the parasite's location to the lamina propria between the vascular endothelium and the alveoli and bronchi (Fig. 2a and Supplementary Fig. S2). Trypanosomes were located in the lung interstitium underneath the endothelium of capillaries, veins and arteries, always in close association with collagen (Fig. 2d, e) and were even found in dense connective tissue as it occurs in the perichondrium of upper bronchi and in the adventitial wall of larger intrapulmonary vessels (Supplementary Fig. S3). Parasites appeared to cluster together forming nests of multiplying long slender parasites as evidenced by the presence of double flagella (Fig. 2b, c). White blood cells were observed in close vicinity of the multiplying trypanosomes (Fig. 2f).

*Trypanosoma brucei* is known to form membranous nanotubes originating from the flagellar membrane, which then dissociate in free extracellular vesicles (EVs)[22]. The lung-resident parasites were shown to secrete EVs (137.98 +/− 23.21 nm, $n = 26$) as beads on a string, which has not been shown before in an in vivo setting (Fig. 2g). Cell–cell communication between the flagellar pockets of two parasites via a string of EVs was also observed (Fig. 2h).

### Naturally transmitted *T. brucei* parasites change the immunological cell composition of the lungs

The above results clearly demonstrate that trypanosomes enter and actively multiply in the lung tissue. Histopathological analysis showed a clearly detectable degree of inflammation at 21 dpi as characterized by the infiltration of numerous immune cells (Fig. 3). Immunofluorescence staining showed the presence of neutrophils (Ly6G+ cells) during infection (Supplementary Fig. S4). The dynamic and quantitative changes in various subsets of innate myeloid and adaptive lymphoid immune cells were determined using flow cytometry. Gating of live (DAPI−) CD45+ cells showed a more than twofold increase in the number of live white blood cells (Fig. 4a, b). During the peak of infection (10 dpi), the main cell types recruited significantly to the lung tissue comprised CD11b+ Ly6C+ monocytes/macrophages ($p < 0.0001$) and CD11b+ CD11c+ dendritic cells ($p < 0.0001$), showing a more than threefold increase in absolute numbers. Other signs of cellular recruitment and/or activation were the increase of F4/80+ macrophages, B220+ IgM− B cells, CD4+/CD25+ αβ T cells and CD3+ γδ T cells. The number of CD11b+ CD11c+ SiglecF+ alveolar macrophages remained stable at the evaluated time points. Most strikingly was the almost 10-fold decrease in the number of CD11b+ CD11c$^{lo/−}$ SiglecF+ eosinophils during acute infection ($p < 0.0001$) as well as the surprisingly low number of infiltrating CD11b+ Ly6G+ neutrophils (Fig. 4c–f and Supplementary Figs. S5, S6), which are known to infiltrate trypanosome-infected tissues such as the skin[23] and adipose[24]. Toward a later stage of infection (21 dpi), a continued infiltration of F4/80+ macrophages, dendritic cells and Ly6C+ monocytes/macrophages was observed with 4-, 8- and 10-fold increases as compared to naive tissue. At 21 dpi, neutrophil levels were increased 6-fold and the level of eosinophils was restored to that of naive animals. (Fig. 4c, d and Supplementary Fig. S5). The lymphoid cellular response was marked by a 3- to 4-fold reduction in the number of B220+ IgM+ B cells and NK cells (Fig. 4e, f and Supplementary Fig. S6).

### Lung infection induces major transcriptional changes

To gain detailed insight into the impact of parasite presence in the lungs, nCounter digital transcriptomics was performed on RNA extracts collected from infected mice at 10 and 21 dpi and corresponding non-infected (NI) littermate controls (data available in the source data xlsx-file). Principle component analysis revealed distant profiles between infected and NI animals, with lung samples collected at 10 and 21 dpi clustering more closely together (Fig. 5a). When considering a false discovery rate of 0.05 and a ≥ 1 log fold change (LogFC), 239 genes were differentially expressed (DE) between infected and NI lung tissue. Although a large proportion of differentially expressed genes ($n = 150$) was shared between 10 and 21 dpi, a

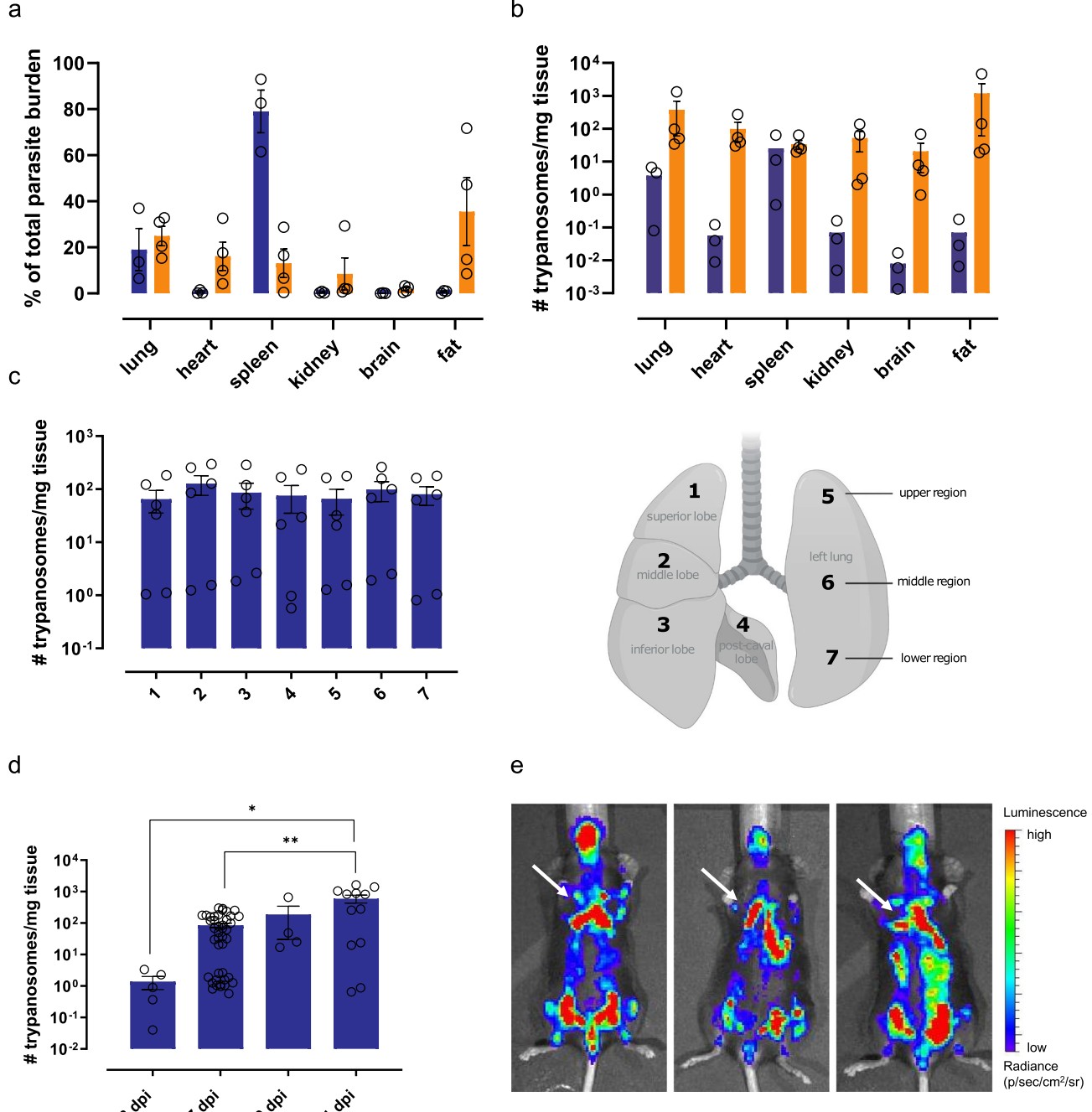

**Fig. 1 | Infection dynamics in *T. b. brucei* infected mice. a–d** C57BL/6JRj mice were infected via the bite of a *T. b. brucei* AnTar1 infected *G. morsitans* tsetse fly. After perfusion of the vascular system, tissues were collected for RNA extraction. Parasite burdens were determined via RT-qPCR targeting the spliced leader (SL) of the parasite. RT-qPCR targeting *Eef2* was performed to normalize for the amount of mouse tissue. **a, b** Parasite burden in the lungs, heart, spleen, kidney, brain, and gonadal adipose tissue of naturally infected mice (3 dpi: *n* = 3, 10 dpi: *n* = 4). **c** Parasite burdens in various segments of the lungs of three independent mice collected at 7 dpi. Absolute parasite numbers were determined via a standard curve. Created with BioRender.com. **d** Parasite burdens in the lungs at different time points during infection (3 dpi: *n* = 3, 7 dpi: *n* = 21, 10 dpi: *n* = 4, 21 dpi: *n* = 6). Statistical comparisons were made using Kruskal–Wallis test (two-sided) with a Dunn's multiple comparisons test. *p = 0.0255, **p = 0.0074. Diagram was adapted from ref. 72. **e** Organ distribution of *T. b. brucei* AnTat1.1E^PpyRE9 parasites determined via bioluminescent imaging at 7 dpi in three different mice. Arrows: bioluminescent signal emitted from the lung tissue. Data are represented as means ± standard error of the mean.

substantial number of transcripts was still differential for the specific infection stage (peak parasitaemia, 10 dpi: *n* = 47; established infection with CNS involvement, 21 dpi: *n* = 42) (Fig. 5b). The clustered heat map of the top 50 DE genes showed that the overall transcriptional profile of the lung tissue was highly affected by infection and with more modest differences between 10 and 21 dpi (Fig. 5c). Hierarchical clustering of these genes resulted in 4 clusters. Cluster S1 was highly

upregulated at 10 and 21 dpi and is annotated by inflammatory responses featuring IFN-γ- and IFN-α-responses, IL-2-, IL-6-, and TNF-signalling and allograft rejection (Fig. 5c, d). The genes included in cluster S2 (*Cxcl13, Ms4a7, Pdcd1lg2, Nos2* and *Ccl8*) were slightly upregulated at 10 dpi during peak parasitaemia and highly upregulated at 21 dpi (Fig. 5c). Cluster S3 and S4, corresponding to the feature 'xenobiotic metabolism', constitute genes that were downregulated

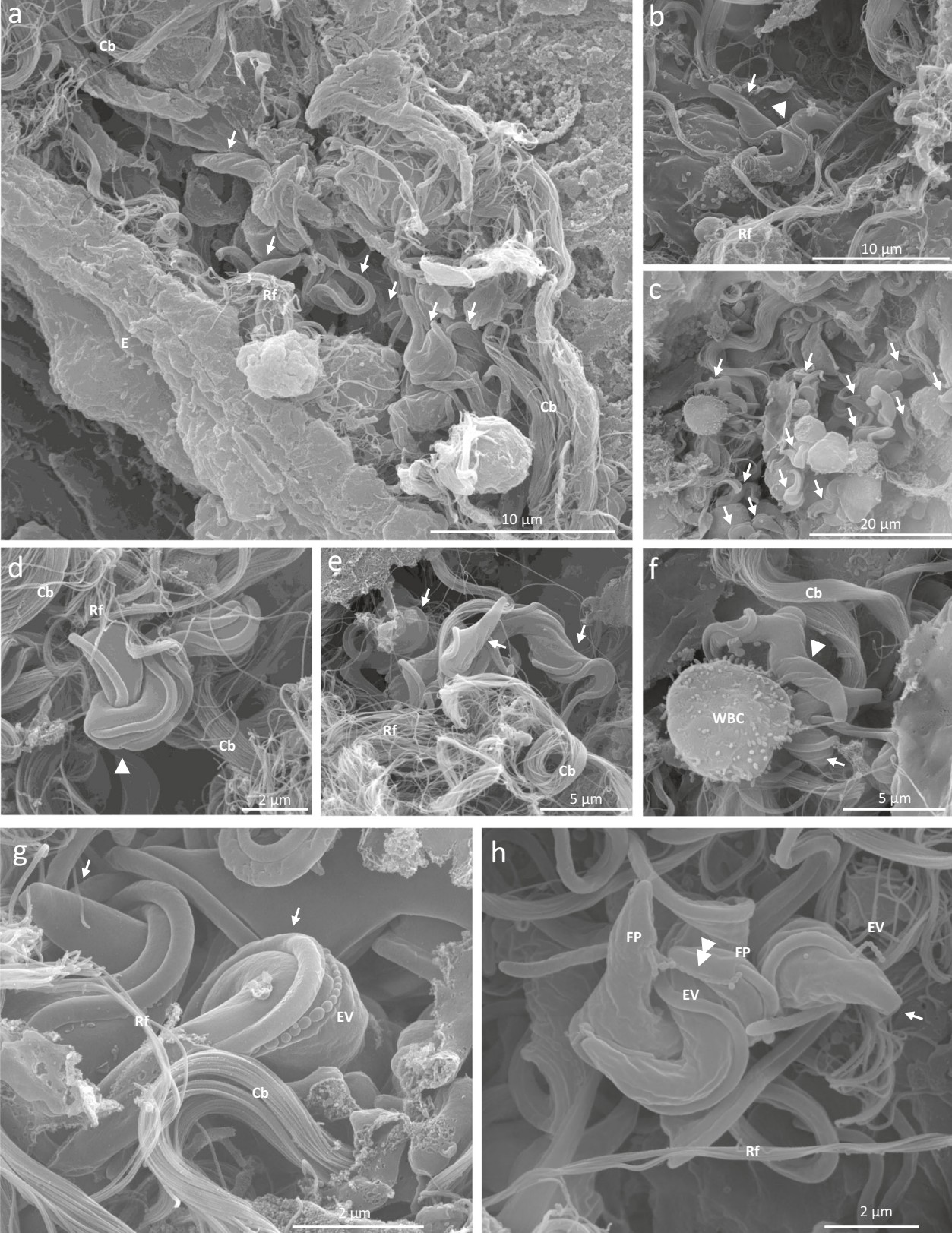

**Fig. 2 | Scanning electron microscopic images of lung tissue of *T. b. brucei* infected mice.** Groups of three C57BL/6JRj mice were infected via the bite of a *T. b. brucei* AnTar1 infected *G. morsitans* tsetse fly. After perfusion of the vascular system at 10 dpi, lungs were fixed in glutaraldehyde and embedded in 1% agarose prior to vibratome sectioning. **a**–**f** Extravascular parasites embedded in lung tissue. **g**–**h** Examples of extracellular vesicles as "beads on a string". Cb collagen bundle, Rf reticular fibre, WBC white blood cell, E endothelium, EV extracellular vesicles, FP flagellar pocket, arrows: *T. b. brucei* parasites, arrowheads: multiplying parasites, double arrowhead: cell–cell communication via string of extracellular vesicles. EVs size evaluation in **h**, 137.98 +/− 23.21 nm (185.64 − 89.93 nm), *n* = 26, not-corrected for depth.

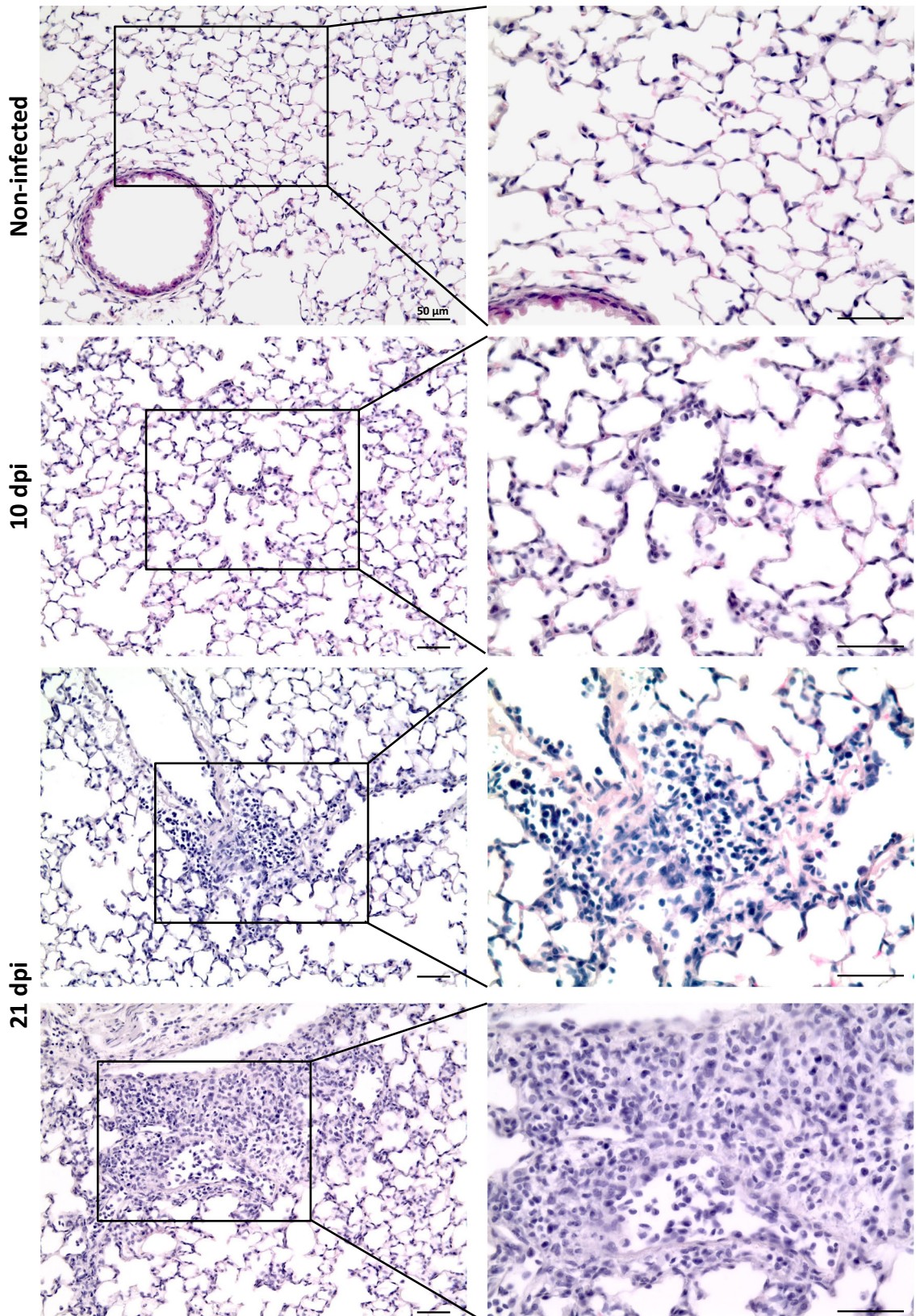

**Fig. 3 | Histopathological analysis of lung tissue of *T. b. brucei* infected mice.** Groups of three C57BL/6JRj mice were infected via the bite of a *T. b. brucei* AnTar1 infected *G. morsitans* tsetse fly. Three mice were processed as non-infected control. After perfusion of the vascular system and fixation of the lungs via intratracheal instillation at 10 and 21 dpi, lungs were collected *en bloc* and the right lobes were embedded in paraffine prior to sectioning. Sections were stained with haematoxylin-eosin. Scale bar size is 50 μm. Similar observations were made in 3 mice per group and representative images are shown.

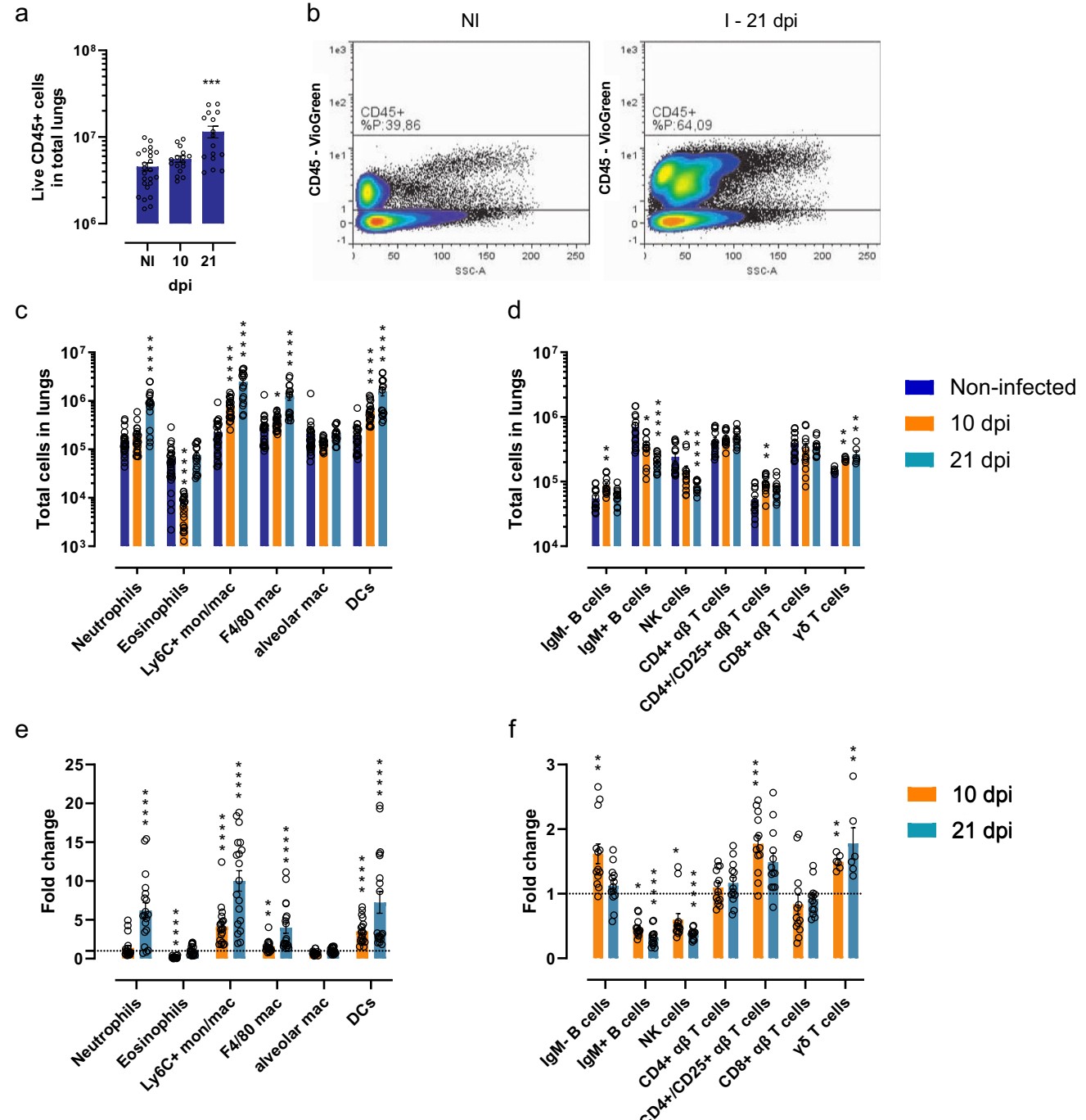

**Fig. 4 | Immune cell composition of lung tissue of *T. b. brucei* infected mice.** Groups of three to five C57BL/6JRj mice were infected via the bite of a *T. b. brucei* AnTar1 infected *G. morsitans* tsetse fly. Three mice were processed as non-infected controls. After perfusion of the vascular system at 10 and 21 dpi, lung tissue was collected and homogenized to obtain single-cell suspensions. The graphs represent **a** the total number of live CD45⁺ cells in the lungs (total number of animals: NI: $n = 12$, 10 dpi: $n = 8$, 21 dpi: $n = 8$ examined over three independent experiments), **b** flow cytometry dot plots illustrating the percentage of CD45⁺ cells in naive (NI) and 21 dpi mice, the total number of **c** myeloid and **d** lymphoid cells in the lungs and **e**, **f** the fold change compared to non-infected control mice calculated based on the absolute cell numbers. **c**, **e** Total number of animals: NI: $n = 15$, 10 dpi: $n = 11$, 21 dpi: $n = 8$ examined over three independent experiments. **d**, **f** Total number of animals: NI: $n = 9$, 10 dpi: $n = 8$, 21 dpi: $n = 8$, γδ T cell: $n = 3$ examined over two independent experiments (except for the γδ T cells, where data are from a single experiment). Mon monocytes, mac: macrophages, DCs dendritic cells. Data are represented as means ± standard error of the mean of at least two independent repeats. Statistical comparisons were made using the Kruskal–Wallis (two-sided) with a Dunn's multiple comparisons test between non-infected and either 10 dpi or 21 dpi. *$p < 0.05$, **$p < 0.01$, ***$p < 0.001$, ****$p < 0.0001$. Exact $p$-values: **a** ***$p = 0.0004$, **c** *$p = 0.0375$, **d** **$p = 0.0024$, *$p = 0.0242$, *$p = 0.0197$, **$p = 0.0015$, **$p = 0.0099$, **$p = 0.0049$, **e** **$p = 0.0066$, **f** **$p = 0.0011$, *$p = 0.0144$, *$p = 0.0379$, ***$p = 0.0006$, **$p = 0.0099$, **$p = 0.0049$.

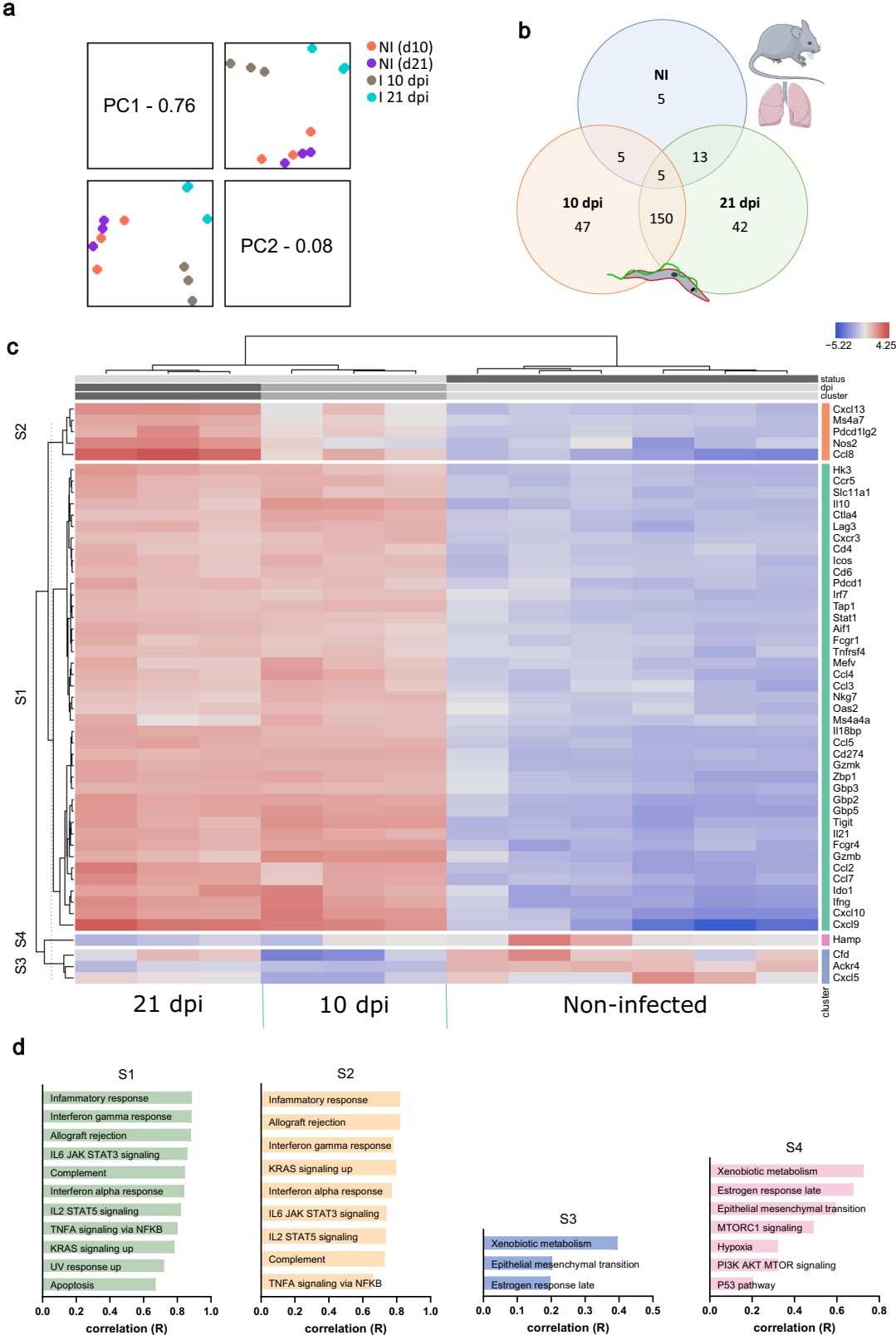

upon infection. *Cfd*, *Ackr4* and *Cxcl5* were mainly downregulated at 10 dpi and *Hamp* mainly at 21 dpi.

The top up- and downregulated genes for the different comparisons (10 dpi vs NI, 21 dpi vs NI and 21 dpi vs 10 dpi) are represented in volcano plots (Fig. 6a–c). The transcriptional changes upon infection were dominated by *Ifng* and its downstream genes (e.g. *Stat1*, *Irf1*, *Fcgr1*, *Nos2*, *Cxcr3*) and IFN-dependent CXCR3 binding chemokines

(e.g. *Cxcl9*, *Cxcl10*). A marked upregulation of genes corresponding to the Th1 signature and pro-inflammatory response was documented (e.g. *Tbx21*, *Eomes*, *Ifng*, *Nos2*, *Stat1*, *Gzmb*, *Gzmk*, *Il21*, *Cxcl9*, *Cxcl10*, *Ccl2*, *Ccl5*, *Ccl8*, *Ccr5*) as described previously for *T. brucei* infected adipose tissue[24]. In contrast, a large upregulation of negative immune checkpoint regulators was observed (e.g. *Tigit*, *Ido1*, *Cd274* (PD-L1), *Lag3* (CD223), *Pdcd1lg2* (PD-L2, CD273), *Ctla4* (CD152)) which are

**Fig. 5 | Transcriptome of *T. b. brucei* infected lung tissue. a** PCA analysis of the nCounter digital transcriptomics data revealing distant clustering between lung tissue samples from infected (I) and non-infected littermate controls (NI) at 10 and 21 dpi. **b** Venn diagram showing the number of overlapping genes between NI (blue), 10 (red) and 21 (green) dpi. **c** Heatmap showing gene expression of the top 50 host gene transcripts quantified by nCounter digital transcriptomics in lung tissue samples of 10 dpi, 21 dpi and NI mice, sorted by two-way hierarchical clustering. Top gene selection is based on standard deviation, LogFC expression across the samples and/or principal components. Expression levels are represented with a colour scale ranging from blue to red (overexpression). **d** Top ranked annotation features (by correlation) for each gene cluster as defined in the heatmap. The top features of the heatmap are divided into gene clusters based on their expression profile patterns. Parts of the figure (panel b) were drawn by using pictures from Servier Medical Art. Servier Medical Art by Servier is licensed under a Creative Commons Attribution 3.0 Unported License (https://creativecommons.org/licenses/by/3.0/). Created using Omics Playground.

involved in the negative regulation of T cell functioning. The regulatory T cell marker *Foxp3* was not differentially regulated. A strong upregulation of *Il10* and *Il21* was noted and a downregulation of *Il12a* and complement-associated genes (e.g. *Hc*, *Cfd*) was observed (Fig. 6d). Flow cytometric analysis of infected lung tissue showed a nearly complete absence of neutrophil influx during the early infection phase (10 dpi) which can be linked to the downregulation of *Cxcl5* and lack of *Cxcl1* and *Cxcl2* upregulation responsible for neutrophil chemotaxis during inflammation[25]. Interestingly, a marked downregulation of B cell receptor-associated genes (e.g. *CD79a*, *CD79b*, *CD19*, *Syk*, *Ms4a1-CD20*) was observed only at 21 dpi, corresponding to the drop in B220⁺ IgM⁺ B cells observed in our immunocytological analyses. Correlation analysis suggests a link between CCL8 and the recruitment of myeloid cells (*Ms4a7*) with antigen-presenting capacity (*H2.Ab1*) and the transcription of B cell chemotactic factor *Cxcl13* (Supplementary Fig. S7b). Interestingly, CXCL13 has been identified as a diagnostic marker in cerebrospinal fluid for stage-II HAT[26]. Correlations were also found between *Cd19* ($R = 0.9134$), *Cd79a* ($R = 0.9865$), *Cd79b* ($R = 0.9620$) and the apoptotic protein *Bik*, which may be involved in the B cell depletion during infection (Supplementary Fig. S7c). Another interesting downregulation was observed at 21 dpi for *Hamp* encoding the antimicrobial protein hepcidin with a role in iron homeostasis (Fig. 6d). During both *T. brucei* and *Plasmodium* infections hepcidin levels are linked to the development of anaemia[27,28]. Observations in *Hamp*⁻/⁻ mice indicate that there is not likely a direct impact on the parasite, but downregulation may increase susceptibility to opportunistic infections given the antibacterial and antifungal activity of hepcidin. A strong upregulation of *Ido1* was also observed throughout the infection, encoding the haem-containing enzyme 2,3-dioxygenase that was previously associated with reduced circulating tryptophan levels and control of parasite burdens[29]. Biomarker analysis identified multiple genes that allow for differentiation between infected and naive animals and even differentiate between the early peak (10 dpi) and more progressed disease stage with CNS involvement (21 dpi) with *Ccl8* and *Cd19* being strong differentiators in opposite directions of the regulation (Supplementary Fig. S7b, c).

Gene ontology (GO) enrichment analysis confirmed our above findings of a negative regulation of T cell activation throughout the infection, B cell receptor signalling only present at 10 dpi and neutrophil chemotaxis and migration at 21 dpi (Figs. 7a, 8a–c). Cellular deconvolution analysis, which estimates the proportion of cells in a tissue based on the gene expression profiles, also pointed to B cell memory depletion upon *T. brucei* infection[30], a strong M1 macrophage polarization with gradual increase of M2 towards 21 dpi, a transient reduction of eosinophils and activation of NK cells at 10 dpi and a strong induction of γδ T cells (Fig. 7b, c) which were previously described to be activated in trypanotolerant but not in susceptible cattle breeds pointing to a role in disease control[31,32]. These cellular fingerprints were conserved over the different applied computational methods for cellular deconvolution (Fig. 7b) and supported by the flow cytometry data (Fig. 4).

### Parasite presence does not detrimentally affect lung function

Mechanical parameters of lung function were assessed using a mouse ventilator (FlexiVent™). During the ventilation measurements, mice were exposed to aerosols with increasing concentrations of the bronchoprovocation agent methacholine. Despite the presence of pulmonary parasites and the observed infiltration of various immune cells and an increase in transcription of inflammatory cytokines, all measured lung mechanical parameters including resistance, elastance, airway resistance, tissue damping and tissue elastance were comparable to those seen in the non-infected control group. Treatment of mice with LPS resulted in significant changes in lung function across all measured parameters (Fig. 9a).

### Parasite presence affects lung co-infections

Given that trypanosome infection and the immunological changes in the lung tissue may influence susceptibility to other pulmonary infections, a co-infection experiment was carried out by intranasal exposure to respiratory syncytial virus (RSV) virus. Despite the small group size, a clear and significant trend was observed ($p < 0.01$). Mice only exposed to the RSV virus showed an expected peak of viral load at day 4 post infection with a reduction by day 8 and the absence of infectious virus in the plaque assay (Fig. 9b, c). The trypanosome-infected group started off with a lower viral load, likely due to prior immune activation, but mice were unable to control RSV infection by day 8 (Fig. 9b, c). Concurrent changes in the lung microbiome and signs of dysbiosis were monitored by a 16S rRNA RT-qPCR (Fig. 9d) and using mouse *Eef2* as a reference gene (Fig. 9e). Compared to the RSV-infected control mice, the lung bacterial load was significantly higher ($p < 0.05$) in the trypanosome-infected group at day 4 post RSV exposure (Fig. 9d).

### Discussion

The number of reported cases of sleeping sickness has rapidly declined due to tremendous control efforts based on vector control, passive and active case finding, and treatment. This incentivised the WHO to set a goal for zero transmission of gHAT by 2030[33,34]. Although this seems within reach, vigilance is needed since there are still important knowledge gaps that may endanger this goal. For gHAT, the question remains where the parasite resides in inter-epidemic periods[35]. This may be partially attributed to asymptomatic carriers who remain undiagnosed due to the lack of symptoms and often undetectable parasite levels in the blood[36]. Trypanosomes were found to prominently accumulate in adipose tissue[13,24] as well as to colonize various other tissues (i.e. testis[37], pancreas[38], spleen[39], cardiac muscle[40] and brain[41]) showing specific adaptations to these local environments. Recent research in experimentally infected mice and a retrospective analysis of human skin biopsies identified the dermis as a previously overlooked reservoir in asymptomatic individuals[9–11]. Skin-resident trypanosomes form a quiescent population that persists in asymptomatic individuals and can be recovered by tsetse flies to continue disease transmission[11,12]. On average, 20% of the skin parasites are stumpy forms previously presumed to be the only insect infectious stages, although recent observations demonstrated that proliferative slender stage parasites can also efficiently infect the tsetse fly vector[42].

Our present research has identified the lungs as a site of parasite proliferation during natural onset and throughout *T. brucei* infection (Fig. 10). Pulmonary parasites are found embedded in the tissue, actively multiplying in the extravascular space and eliciting an immunological response with limited consequences on physiological function based on a gold standard for in vivo mechanical lung function

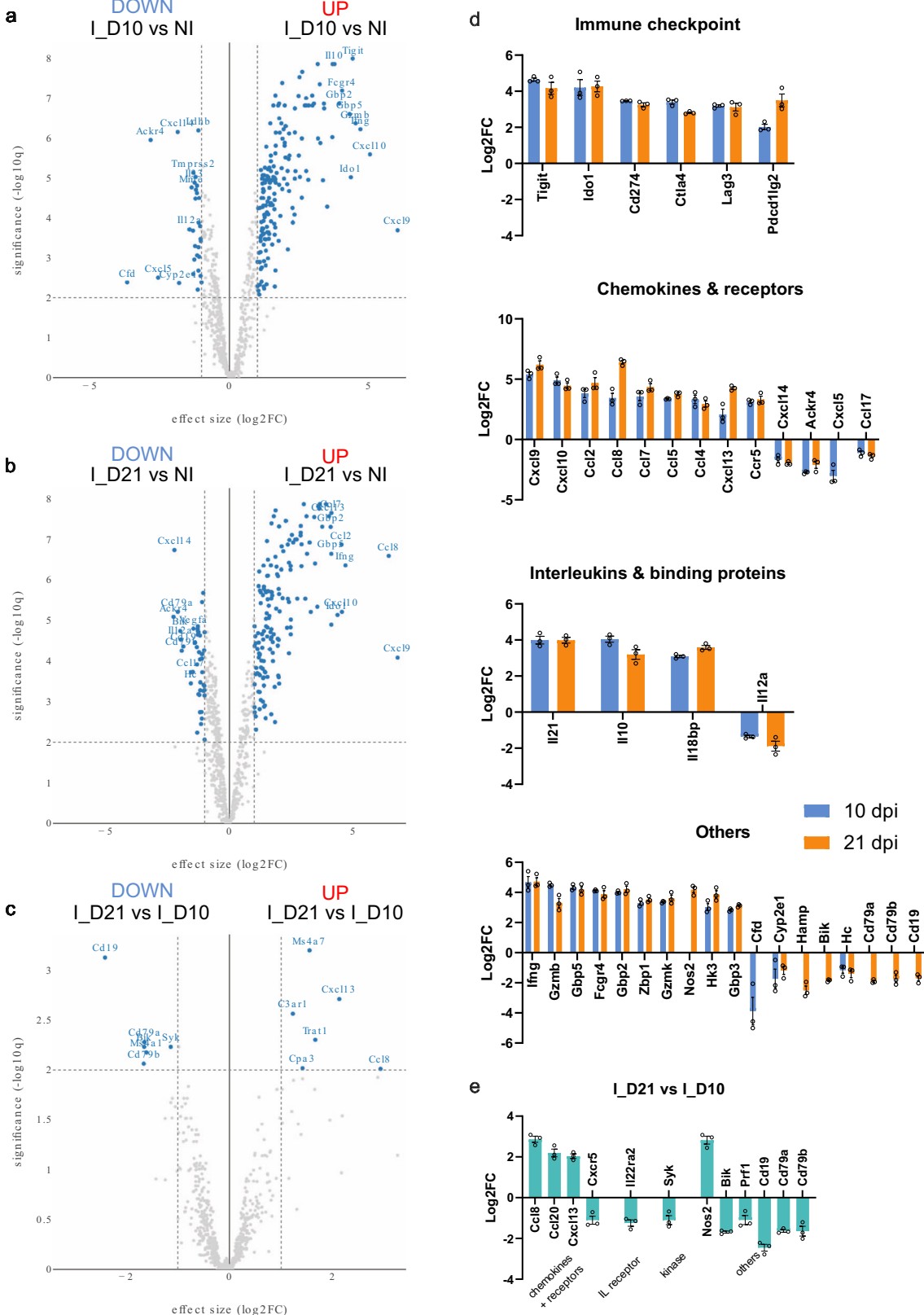

**Fig. 6 | Detailed differential gene expression of *T. b. brucei* infected lung tissue.** **a**–**c** Volcano plots of differentially expressed (DE) genes plotting fold-change (*x*-axis) versus significance (*y*-axis). Volcano plot of DE genes of **a** 10 dpi (*n* = 3) versus NI (*n* = 6) mice, **b** 21 dpi (*n* = 3) versus NI (*n* = 6) mice, **c** 21 dpi (*n* = 3) versus 10 dpi (*n* = 3) mice. **d** Overview of the DE genes with the largest fold change (Log2FC > 3, Log2FC < −1.5) in different biological categories. Graphs represent the Log2FC ± SEM of gene expression in lung tissue at 10 (*n* = 3) and 21 dpi (*n* = 3) compared to NI (*n* = 6) mice. **e** Overview of the DE genes between 21 dpi (*n* = 3) and 10 dpi (*n* = 3) mice, with the largest fold change (Log2FC > 2, Log2FC < −1). Graphs represent the Log2FC ± SEM of gene expression between 21 dpi and 10 dpi. Created using Omics Playground.

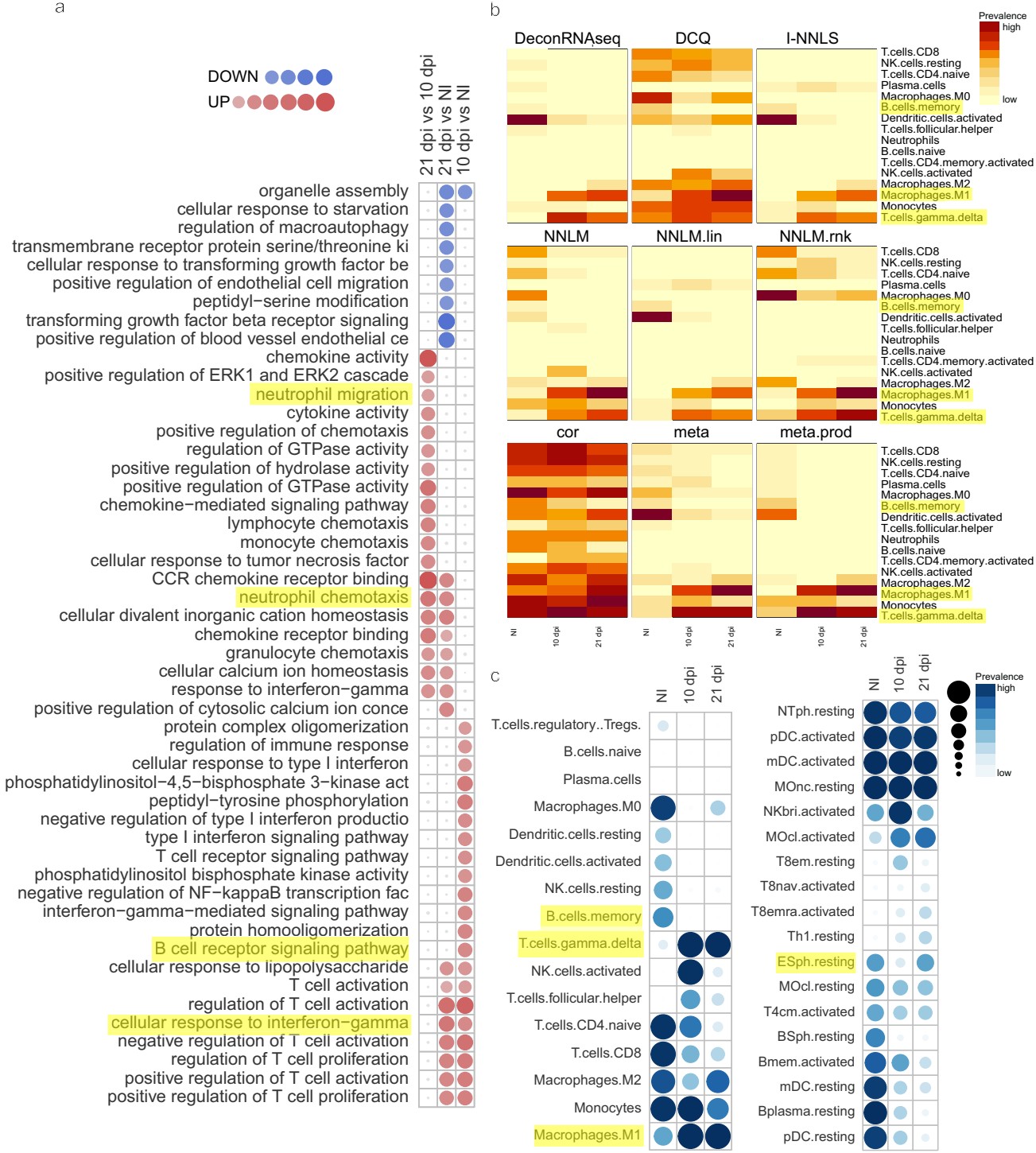

**Fig. 7 | Gene ontology analysis and cell type mapping of *T. b. brucei* infected lung tissue. a** Activation matrix visualizing the enrichment of GO terms across the different comparisons (21 dpi vs 10 dpi, 21 dpi vs NI, 10 dpi vs NI). Red colour corresponds to upregulation and blue colour corresponds to downregulation of the corresponding pathway. Circle size corresponds to the number of genes involved in the specific pathway. **b** Heatmap visualizing the distribution of inferred immune cell types based on nCounter digital transcriptomics for different computational deconvolution methods (DeconRNAseq, DCQ, I-NNLS, NNLM, NNLM.lin,

NNLM.rnk, cor, meta, meta.prod) in the Omics Playground software. **c** Overview dot plot of the cell type mapping. Some major changes are highlighted in yellow. NTph neutrophil, pDC plasmacytoid dendritic cell, mDC myeloid dendritic cell, MOnc non-classical (CD14$^{lo}$ CD16$^{+}$) monocyte, NKbri activated natural killer cell, T8em effector memory CD8$^{+}$ T cell, T8nav naive CD8$^{+}$ T cell, T8emra CD45RA$^{+}$ effector memory CD8$^{+}$ T cell, ESph eosinophil, MOcl classical (CD14$^{hi}$ CD16$^{-}$) monocyte, T4cm central memory CD4$^{+}$ T cell, BSph basophil, Bmem memory B cell, Bplasma plasma B cell. Created using Omics Playground.

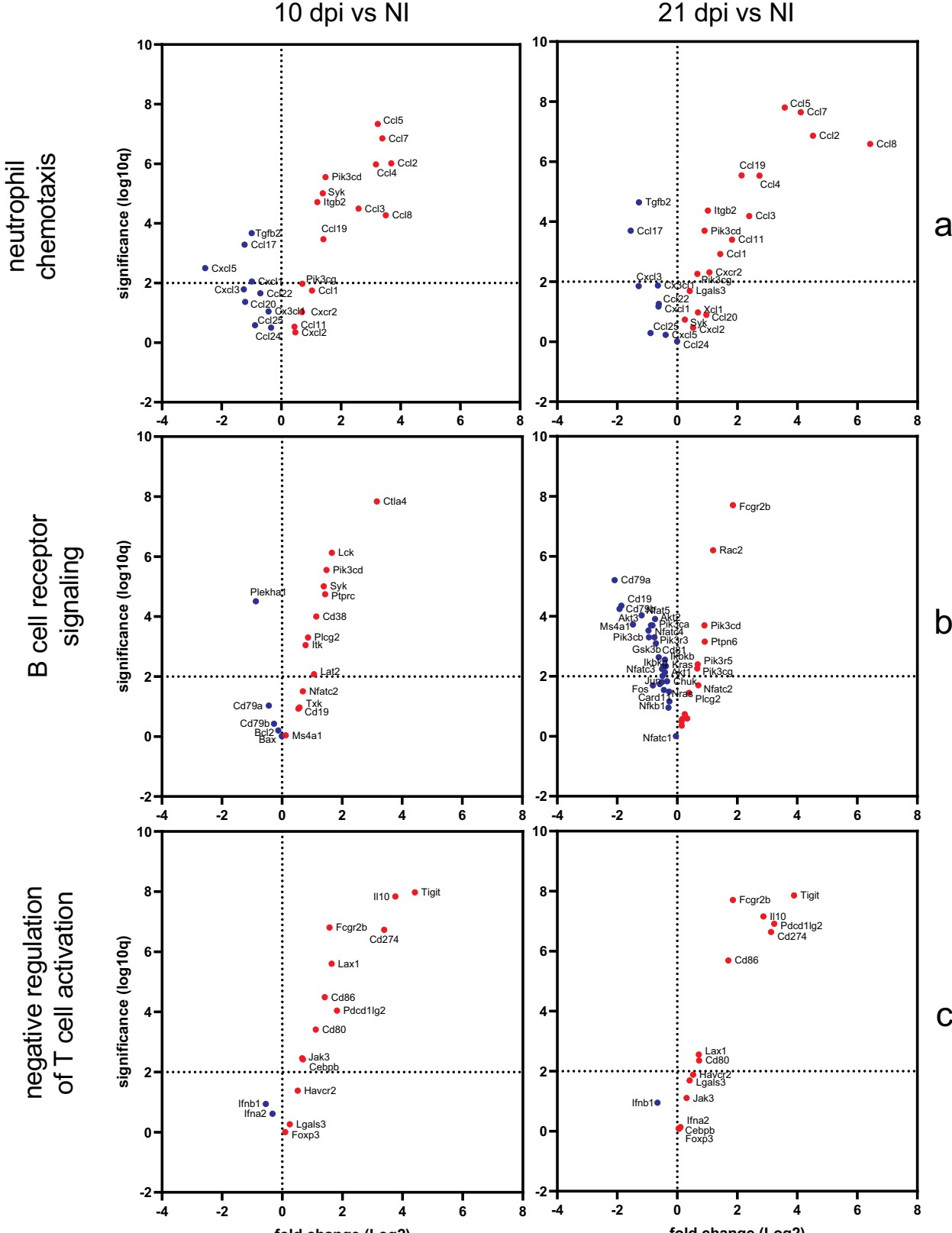

**Fig. 8 | Gene ontology analysis of *T. b. brucei* infected lung tissue.** Volcano plot of DE genes involved in the GO terms: **a** neutrophil chemotaxis, **b** B cell receptor signalling pathway and **c** negative regulation of T cell activation.

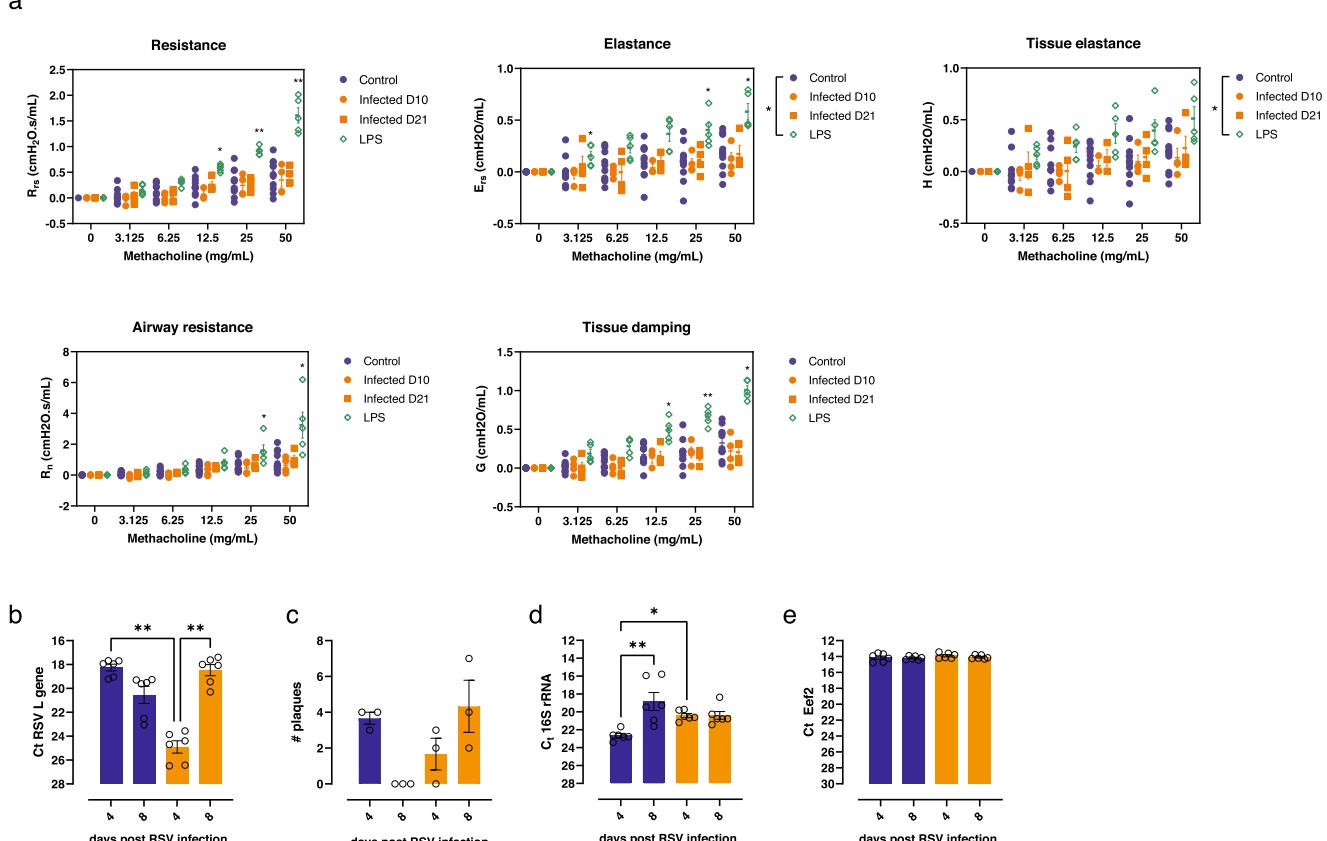

**Fig. 9 | Evaluation of lung functioning and susceptibility to opportunistic infections of *T. b. brucei* infected mice. a** Groups of at least four C57BL/6JRj mice were infected via the bite of a *T. b. brucei* AnTar1 infected *G. morsitans* tsetse fly. After sedation using pentobarbital at 10 and 21 dpi, mice were intubated and connected to the FlexiVent™ system. Lung parameters were measured upon inhalation of increasing doses of methacholine. Data are represented as means ± standard error of the mean of two independent repeats. Total number of mice: control (*n* = 11), 10 dpi (*n* = 4), 21 dpi (*n* = 4) and LPS (*n* = 5). Statistical comparisons were made using the Kruskal–Wallis test. \**p* < 0.05, \*\**p* < 0.01. **b**–**e** Mice were randomly allocated to two groups (*n* = 6) of which one was infected with *T. b. brucei* AnTat1.1E^PpyRE9 via the bites of infected tsetse flies. The other group served as non-infected control (NI). 13 days after the tsetse bite, mice of both groups were

infected intranasally with RSV type A and sacrificed at days 4 and 8 post RSV infection. Lung tissue was collected for RNA extraction and to prepare a tissue homogenate for a viral plaque assay. **b** Viral load was determined by RT-qPCR targeting the RSV type A *L* gene showing substantial changes in the viral infection kinetics. Exact *p*-values: 0.001 and 0.0049. **c** A plaque assay was performed to determine the amount of live virus in the lung tissue. These data mirror the changes observed by RT-qPCR in (**b**). **d** In the same mice, the impact of infection on bacterial dysbiosis was explored by 16S rRNA RT-qPCR. Exact *p*-values: 0.0036 and 0.0256. **e** Corresponding threshold cycle (Ct) values are shown for the *Eef2* reference gene. Statistical comparisons were made using Kruskal–Wallis test (two-sided) with a Dunn's multiple comparisons test. \**p* < 0.05, \*\**p* < 0.01.

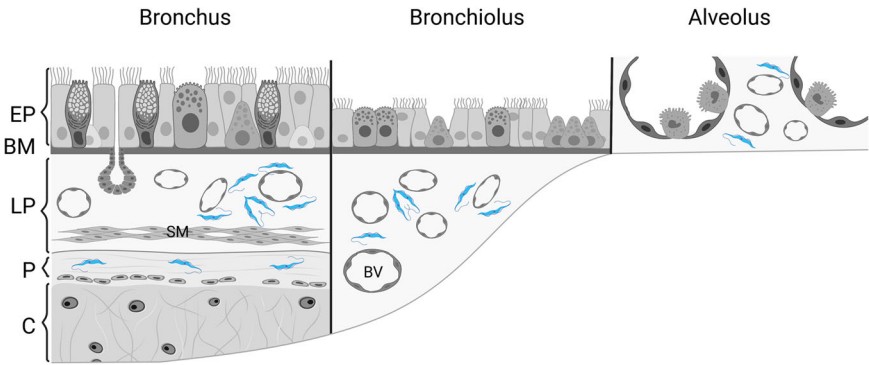

**Fig. 10 | Parasite location in the lung.** Schematic representation of the anatomical structure of the human airway (bronchus, bronchiolus and alveolus) with the locations where African trypanosomes (blue) are expected to be found based on

the performed mouse infection studies. EP epithelium, BM basal membrane, LP lamina propria, P perichondrium, C cartilage, BV blood vessel. The figure was adapted from ref. 73 and created with BioRender.com.

measurements. This phenomenon was previously described in the skin where the pathological consequences also remained limited despite high parasite burdens[9]. The limited organ-specific pathology may be the underlying reason for the only recent appreciation of the relative importance of the skin and lungs as target organs[10,18]. Although 20% of HAT patients suffer from respiratory symptoms, the role of the pulmonary parasites tropism remains largely unexplored[18].

Scanning electron microscopy demonstrated a parasite localization in the lamina propria surrounding blood vessels of the alveoli and bronchi. The close vicinity to the endothelium likely allows for nutrient uptake from the blood by transcytosis by endothelial cells[43]. The high oxygen tension may affect metabolic characteristics of the pulmonary parasites and underly circadian differences, as described for microfilariae[44,45]. Trypanosomes appeared embedded in reticular fibres and collagen bundles, an affinity described in multiple tissues including the cardiac atria, skin[11,20] and dermal ulcer (referred to as a chancre) at the infectious bite site[46]. Pulmonary parasites cluster together in nests and release EVs that may participate in cell–cell communication[22] which, to our knowledge, has not yet been documented in an in vivo setting. Similar ultrastructural studies in infected skin[11] could not reveal the relatively frequent EV release as beads on a string observed in the lungs and in vitro[22,47]. Recurrently these EVs were observed at the boundary of the flagellum and the cell body, and frequently, originating at the entry of the flagellar pocket. EVs have multiple functions and can be involved in parasite communication and in the interaction with immune cells. Fusion of these vesicles with neighbouring parasites allows for the transfer of lipids and proteins contributing to parasite virulence[48]. Intercellular communication between parasites can even transfer serum resistance factors from human pathogenic to non-pathogenic strains[22]. The release of EVs also modulates the host immune response in favour of trypanosome replication[49–51]. The biological importance of the extensive EV release and metabolic adaptations to the high oxygen tension in the lung remains to be understood.

Parasite presence in the lungs coincided with a pronounced inflammatory response reflected by the influx of white blood cells as shown by histopathological and flow cytometric analyses. In general, the high number of immune cells attracted to the lung tissue upon infection does not seem to hamper parasite replication, as evidenced by numerous parasites with double flagella. A detailed study into the immune cell subsets marked an increase in neutrophils and Ly6C$^+$ monocytes, which has been described previously in skin and spleen[23,39]. Although neutrophils are known to be rapidly recruited and monocytes display a more gradual increase[23], a prominent neutrophil influx in the lungs was not observed during the acute infection stage (10 dpi) but appeared only later in the infection (21 dpi). This was confirmed by GO analysis of lung transcript data where neutrophil chemotaxis and migration were only apparent at 21 dpi. Neutrophils play a substantial role in many respiratory diseases ranging from asthma to bacterial/fungal pneumonia and lung cancer, and are often associated with pathology[52]. Despite the substantial influx of neutrophils by 21 dpi, no functional deficits were observed in our *T. brucei* infection model using state-of-the art in vivo lung function measurements. Further research into the local activation and impact of neutrophils is warranted given their previously reported role in promoting early infection[23]. At 10 dpi, a surprising drop in the number of eosinophils was observed and evaluation of the dead cell population (DAPI$^+$) could not account for the nearly complete absence of this cell type. Cell type mapping based on transcriptional data of infected lung tissue also showed a similar decrease of eosinophils at day 10. Eosinophils have been extensively studied in helminth infections[53] and their importance in protecting against fungal infestation of the lungs has been documented[54]. In addition, eosinopenia has been previously shown to be a common feature of acute inflammation and bloodstream infections[55,56]. Also in COVID-19 patients eosinopenia is

frequently observed and confers a higher risk for developing severe disease and for requiring admission to the intensive care unit[57]. Whether eosinophils play a major role during a *T. brucei* infection is still unknown. Previous research has identified two subsets of eosinophils in the lungs: resident and inflammatory eosinophils. Mice lacking resident eosinophils showed an increase in Th2 cell responses[58] which may promote parasite expansion with low levels of pathogenicity[59]. By 21 dpi, eosinophils were replenished to normal levels.

Analysis of the transcriptome of *T. brucei* infected lung tissue showed a response dominated by IFN-γ and a Th1 proinflammatory response, as described previously for other tissues[24]. In addition, transcript analysis pointed into the direction of an overall induction of γδ T cells previously described to play a role in disease control in trypanotolerant animals[31,32]. Cellular deconvolution provides very strong support (*p*-values $<10^{-4}$ to $<10^{-6}$) for the induction of γδ T cells, a cell type known to regulate bacterial clearance in infected lungs via the production of IL-17 and neutrophil recruitment and to protect against lung fibrosis through IL-22[60]. In our lung transcriptome data, *Il17* and *IL22* were not found to be upregulated upon trypanosome infection. Cellular immunoprofiling confirmed a significant increase of γδ T cell numbers alongside CD4$^+$ CD25$^+$ αβ T cells. As this increase of CD4$^+$ CD25$^+$ αβ T cells was not accompanied by *Foxp3* upregulation, this indicates the elevated presence of activated T cells rather than regulatory T cells. The significant upregulation of several immune checkpoint genes may contribute to keeping T cell activation and proliferation in check and to T cell suppression, a hallmark of trypanosomiasis. This regulation may be important to control inflammation and tissue injury, as accumulation of CXCR6$^+$ CD4$^+$ T lymphocytes is pivotal in inducing liver pathology and early mortality[61]. The role in lung pathology remains to be understood. Cellular deconvolution analysis showed a clear M1 macrophage polarization upon infection required for initial parasite control with a gradual increase of M2 upon disease progression by the induction of IL-10 to protect against tissue damage[16]. Also a temporary reduction of eosinophils and activation of NK cells was observed during early infection. In the lymphoid fraction, a drop in numbers of NK cells and B220$^+$ IgM$^+$ B cells was documented. A similar reduction of NK1.1$^+$ cells was observed in the spleen of *T. brucei* infected mice[39]. The decline of B220$^+$ IgM$^+$ B cells in combination with reduced transcription of several B cell receptor associated genes suggests depletion of the pulmonary B cell compartment by the parasite. Cellular deconvolution further pinpoints the effects to a reduction of memory B cells. A rapid depletion of B cells has been extensively demonstrated in other tissues during *T. brucei* infection[30] and recently also for *T. evansi*[62]. Such effects are likely to affect susceptibility to (vaccine-preventable) opportunistic pathogens[30]. A co-infection experiment with RSV showed that trypanosome infection modifies viral infection kinetics, resulting in high viral levels at timepoint where control mice have cleared the infection. Also temporal differences in total bacterial loads were observed in the lung co-infection model. These experimental observations support the concept that trypanosomes may influence susceptibility to viral or bacterial lung infections.

In summary, the study results define the lungs as important parasite reservoir tissue for *T. brucei*. Trypanosomes can be found in extravascular parasite nests and interestingly show EV secretion and prominent interactions with collagen. The high lung burdens may account for the respiratory symptoms that occur in about twenty percent of the patients. Surprisingly, the pulmonary inflammation in the *T. brucei* mouse model did not result in infection-associated pulmonary dysfunction, which may mirror the more common asymptomatic lung infection in patients. Substantially reduced numbers of eosinophils, NK cells and B cells may nevertheless render individuals more susceptible to opportunistic infections. These findings will trigger further studies into the parasitological adaptations and impact of the observed immunological changes in the lungs during

trypanosomiasis. The prompt and permanent colonization of this tissue with high trypanosome levels, may also stimulate exploration of new diagnostic approaches to identify asymptomatic carriers.

## Methods

### Ethics Statement

The use of laboratory rodents was carried out in strict accordance with all mandatory guidelines (EU directives, including the Revised Directive 2010/63/EU on the Protection of Animals used for Scientific Purposes that came into force on 01/01/2013, and the declaration of Helsinki in its latest version) and was approved by the Ethical Committee of the University of Antwerp, Belgium [UA-ECD 2017–04 and 2022–55]. Tsetse fly pupae were imported from Bratislava via a collaboration with Dr. Peter Takac (Slovak Academy of Sciences, Slovakia).

### Animals and parasites

Female C57BL/6JRj mice (*Mus musculus*, 6–8 weeks) were purchased from Janvier (Le Genest-Saint-Isle, France) and were used for all in vivo experiments. Food for laboratory rodents (Carfil, Arendonk, Belgium) and drinking water were available *ad libitum*. Animals were housed in an individually ventilated caging system controlled at 22–26 °C, 40–60% relative humidity and a 12 h alternating dark/light cycle. All housing standards specified in Annex 4 of the Belgian Royal Decree of 29 May 2013 were respected. The animals were kept in quarantine for at least 5 days before infection and were randomly allocated to the experimental units.

*Trypanosoma brucei brucei* AnTAR1 and two transgenic strains of the pleomorphic *T. b. brucei* AnTat1.1E expressing the red fluorescent protein of a *Discosoma* coral (DsRED, Clontech)[11] or a red-shifted firefly luciferase (PpyRE9)[23] were used for flow cytometry and bioluminescent imaging, respectively [kind gifts of Nick Van Reet and Philippe Büscher (Institute of Tropical Medicine, Antwerp, Belgium)].

### Establishment of infection in the tsetse fly

*Trypanosoma b. brucei* AnTat1.1E^PpyRE9/DsRED parasites were used to infect tsetse flies (*Glossina morsitans morsitans*) by feeding newly emerged flies with an infected blood meal supplemented with 10 mM reduced L-glutathione (Sigma-Aldrich, G6013). The infected blood meal consisted of a mixture of parasitized blood from infected mice at 5–7 days post-infection (dpi) and defibrinated horse blood to obtain a concentration of >10^6 BSF parasites/mL. After this infected blood meal, tsetse flies were fed every 2–3 days with uninfected defibrinated horse blood. At 4 weeks post-infection (wpi), the tsetse flies were allowed to probe on pre-heated glass slides, which were then microscopically examined for the presence of metacyclic trypanosomes. Establishment of natural infections in mice.

### Establishment of natural infections in mice

Tsetse flies harbouring metacyclic parasites in the salivary glands were used for natural transmission. Mice were anesthetized with ketamine (100 mg/kg) and xylazine (20 mg/kg) and individual flies (1 fly per mouse) were allowed to bite on the ear dermis. Parasitaemia was monitored by microscopic counting of diluted tail vein blood using a Neubauer-improved haemocytometer (Supplementary Fig. S8).

### In vivo bioluminescent imaging

Parasite burdens were evaluated by in vivo bioluminescent imaging (BLI) following an infective bite with *T. b. brucei* AnTat1.1E^PpyRE9 on the left ear. BLI was performed 3 minutes after intraperitoneal (IP) injection of 15 mg/kg D-Luciferin (Beetle Luciferin Potassium Salt, Promega, E1601) with the IVIS® Spectrum In Vivo Imaging System (Perkin Elmer) under 2% isoflurane inhalation anaesthesia. Luminescence acquisition was performed over a 180, 60, 5 and 1 s exposure period from the dorsal and ventral side of the animal. Over a period of 14 days, the evolution of parasite burdens was assessed as emitted luminescence

by image analysis using LivingImage v4.3.1 software within a defined region of interest (ROI).

### Isolation of infected lung tissue

After IP injection of an overdose of ketamine (300 mg/kg) and xylazine (60 mg/kg), the abdominal wall and thorax of the mice were opened to expose the heart. The mice were perfused at 100 ml/h using a multi-channel syringe pump (ProSense B.V.) via the right ventricle using Krebs-Henseleit buffer (Sigma-Aldrich, K3753) containing 10 U/mL heparin (Sigma-Aldrich, H4784) to rinse away the blood.

Lung tissue was processed for several purposes, i.e., real-time quantitative PCR (RT-qPCR), nCounter digital transcriptomics profiling, flow cytometry-based immune-profiling, scanning electron microscopy, immunofluorescence microscopy and histopathological staining.

### RNA extraction and RT-qPCR

At 3, 7, 10 and 21 dpi, lung tissue was collected for RNA extraction. Tissues were stored in RNAlater (Sigma-Aldrich, R0901) and processed using the RNeasy plus mini kit (Qiagen, 74134) according to the manufacturer's recommendations. Parasite levels were determined using RT-qPCR with specific primers targeting SL-RNA (F: AACTAACGC-TATTATTAGAA, R: CAATATAGTACAGAAACTG) using the SensiFAST™ SYBR® Hi-ROX One-Step Kit (Bioline, BIO-73005). PCR program: 10 min at 45 °C, 10 min at 95 °C, 40 cycles (15 s at 94 °C, 15 s at 50 °C and 15 s at 60 °C) and a melt curve (15 s at 95 °C, 1 min at 45 °C and 15 s at 95 °C)[63]. To prepare a standard curve, perfused lung tissue of non-infected mice was mixed in RLT lysis buffer with 50 μL of a known number of parasites (range 10^6–10^1). The spiked tissue was further processed following the manufacturer's instructions. RT-qPCR targeting *eukaryotic translation elongation factor 2* (*Eef2*), a mouse reference gene, was performed in parallel to normalize between tested samples[64]. The parasite burdens were determined on tissue pieces collected from different lung lobes.

### NanoString digital transcriptomics and bioinformatic analyses

RNA from lung tissue (triplicates for both infected and uninfected conditions at 10 and 21 dpi for nCounter (NanoString) analysis on a nCounter® MAX Analysis System) was extracted as described above. RNA extracts were hybridized to 800 unique capture/reporter pairs (50 bp each) targeting 785 immune transcripts and 12 housekeeping genes, as defined in the Murine Host Response nCounter® panel, as well as six positive and eight negative control probes (all from Nano-String). Results were sequentially corrected for background (negative control probes), technical variation (positive control probes) and RNA content (housekeeping genes) using nSolver 4.0 (NanoString), followed by differential expression and pathway enrichment analysis using OmicsPlayground (by BigOmics Analytics)[65]. Cellular deconvolution of the transcript data was based on various computational algorithms that use a variety of statistical frameworks (DeconRNAseq, DCQ, I-NNLS, NNLM, NNLM.lin, NNLM.rnk, cor, meta, meta.prod).

### Histopathology and immunofluorescence microscopy

After perfusion, the lungs were fixed using 4% paraformaldehyde (Sigma-Aldrich, P6148) in PBS via intratracheal instillation. A canula was inserted in the trachea and the lungs were filled with fixative until they re-occupied the thoracic cavity. Lung tissue was collected *en bloc* together with the heart and oesophagus and fixed for 2 h at room temperature (RT). Deaeration of the lungs was performed in a vacuum chamber. Upon fixation, the tissue blocks were rinsed three times for 15 min in PBS. The right lung lobes were stored in PBS with 0.05% NaN₃ at 4 °C until further processing for paraffin embedding and histopathological analysis. The left lung lobes were stored overnight at 4 °C in 20% sucrose/PBS and processed for cryosectioning and fluorescence microscopy.

For histopathological analysis, 5 μm sections of the right lung lobes were stained with haematoxylin & eosin (HE) and evaluated using an Olympus Bx41 light microscope equipped with a Leica EC3 camera. For fluorescence microscopy, the left lung lobes were embedded in PELCO cryo-embedding compound (Ted Pella, Inc., Redding, CA) and 25 μm thick cryosections were stained with a polyclonal rabbit anti-VSG IgG (1/2500; kindly donated by Benoît Stijlemans, VUB) and a rat anti-mouse CD31 IgG (1/50; ab56299; Abcam) or rat anti-mouse Ly-6G IgG (1/2000; BE0075-1; BioXCell). Sections were pre-incubated with 0.05% thimerosal, 0.01% NaN$_3$, 0.1% BSA, 1% Triton X-100 and 10% normal horse serum in PBS for 1 h. After overnight incubation with the primary antibodies, sections were rinsed and incubated for 4 h with Cy3-conjugated goat anti-rabbit Fab fragments (1/2000) or Cy5-conjugated donkey anti-rat IgG (1/200; Jackson ImmnoResearch Europe Ltd, Suffolk, UK) as secondary antibodies. Anti-VSG and Ly-6G stained sections were counterstained with 5 μg/mL DAPI for 5 min. Finally, sections were coverslipped with Citifluor AF1 mounting solution (Electron Microscopy Sciences, Hatfield, PA). Imaging was performed on a Nikon Eclipse Ti-E inverted microscope attached to a microlens-enhanced dual spinning disk confocal system (UltraVIEW VoX; PerkinElmer, Zaventem, Belgium) equipped with 405, 561 and 641 nm diode lasers for excitation of DAPI, Cy3 and Cy5, respectively.

## Scanning electron microscopy

After perfusion, the lung lobes were subdivided in smaller tissue blocks (5 × 5 × 5 mm) and were fixed overnight at 4 °C using 0.1 M sodium cacodylate buffer containing 2.5% EM-grade glutaraldehyde. The samples were washed 4 × 15 min in 0.1 M sodium cacodylate buffer (pH 7.2). The fixed tissues were next embedded in 1% agarose and cut into 500 μm sections with a Vibratome (Leica). Sections were incubated overnight at 4 °C in 2.5% glutaraldehyde, 0.1 M cacodylate buffer (pH 7.2), and post-fixed in 2% OsO4 in the same buffer. After serial dehydration, samples were critical point dried and coated with platinum by standard procedures. Observations were made in a FEG-ESEM QUANTA F200 (FEI), using secondary electron imaging at 20–30 kV and a working distance of 8–11 mm. Images were analysed (measurements) by the AnalySIS iTEM software (Olympus).

## Preparation of single-cell suspensions

To generate single-cell suspensions from perfused lungs, the Mouse Lung Dissociation Kit (Miltenyi Biotec, 130-095-927) was used. Briefly, resected lung lobes were transferred into gentleMACS™ C tubes (Miltenyi Biotec) containing 2.4 mL of the enzyme mix (Miltenyi Biotec). After running a first lung-specific gentleMACS™ program, samples were incubated for 30 min in a 37 °C water bath, whilst shaken every 5 min, followed by a second lung-specific gentleMACS™ program. Next, cell suspensions were transferred into Falcon tubes and centrifuged for 10 min at 300 × g (4 °C), and pellets were resuspended in 3 mL ammonium-chloride-potassium (ACK) buffer (0.15 M NH$_4$Cl, 1.0 mM KHCO$_3$, 0.1 mM Na$_2$EDTA) for a 7 min erythrocyte lysis at RT. Pellets were recovered after a centrifugation step at 300 × g (4 °C), resuspended in PBS + 0.2% bovine serum albumin (BSA) buffer and filtered through a 100 μm filter (Miltenyi Biotec). Cell suspensions were counted using a KOVA® counting chamber.

## Flow cytometry

Cell suspensions (2 × 10$^7$/mL) were treated with FcγR-blocking agent (anti-CD16/32, clone 2.4G2, BD Biosciences) for 15 min in PBS + 0.2% BSA buffer. Next, cells were incubated for 20 min at 4 °C with a mix of fluorescent conjugated anti-mouse antibodies at optimized concentrations (Supplementary Table S1). DAPI Staining Solution (Miltenyi Biotec, 130-111-570) was used to assess viability. Cells were measured by flow cytometry using a MACSQuant® Analyzer 10 (Miltenyi Biotec) and analyses were performed using FlowLogic™ Software (Miltenyi

Biotec) following specific gating strategies (Supplementary Table S2, Supplementary Figs. S9–11) based on published methods[66–69].

## Lung function analysis

The lung function was determined at 10 and 21 dpi using a FlexiVent™ system (SCIREQ). As a control, naïve mice were instilled intratracheally with 10 μg LPS in 50 μL PBS and subjected to lung function measurements 48 h later. In short, mice were sedated with 150 mg/kg pentobarbital, intubated with a 19 G blunt tip canula and attached to the FlexiVent™ ventilator. Mice were challenged via nebulization with an increasing dose of methacholine (0–50 mg/mL in PBS). Pre-set ventilation protocols of the FlexiVent™ apparatus were applied to characterize resistance, elastance, airway resistance, tissue damping and tissue elastance.

## Impact of *T. brucei* on pulmonary co-infections

Mice were divided into two groups (n = 6). One group was infected with *T. b. brucei* AnTat1.1E$^{PpyRE9}$ via a bite of an infected tsetse. The other group served as non-infected control group. At 13 days post trypanosome infection, mice were infected with human RSV A2 (obtained from BEI resources, NIAID, NIH) via intranasal administration with 10$^6$ PFU/mouse in 75 μL Hank's balanced salt solution (HBSS). At day 4 and 8 post RSV infection, 20 μg of lung tissue was collected for RNA extraction and the remaining was homogenized in HBSS using TissueRuptor Disposable Probes (Qiagen) and centrifuged at 1000 × g for 15 min. The supernatant was stored at −80 °C. RNA was extracted using the RNeasy plus mini kit (Qiagen) according to the manufacturer's recommendations. RT-qPCR was performed using the SuperScript III Platinum One-step Quantitative Kit (Invitrogen, 11732088) with primers specific for the L-gene of RSV type A (F: GCT CTT AGC AAA GTC AAG TTG AAT GA, R: TGC TCC GTT GGA TGG TGT AAT) and a nucleotide probe (5HEX/ACA CTC AAC/ZEN/AAA GAT CAA CTT CTG TCA TCC AGC/3IABkFQ). PCR program: 30 min at 50 °C, 5 min at 94 °C and 45 cycles (15 s at 94 °C, 1 min at 55 °C). In order to assess the effects of infection on bacterial dysbiosis, bacterial loads in the lungs were determined using the SensiFAST™ SYBR® Hi-ROX One-Step Kit (Bioline) and primers specific for 16S rRNA (63F: GCAGGCCTAACACATGCAAGTC, 335R: CTGCTGCCTCCCGTAGGAGT). PCR program: 10 min at 45 °C, 15 min at 95 °C, 40 cycles (15 s at 95 °C and 1 min at 60 °C) and melt curve (15 s at 95 °C, 1 min at 65 °C and 15 s at 95 °C)[70].

The lung tissue supernatant was subjected to an immunodetection plaque assay in HEp-2 cells (obtained via ATCC, CCL-23™) using palivizumab as primary antibody and a horseradish peroxidase-labelled goat anti-human secondary antibody with chloronaphtol for detection[71]. In brief, Hep-2 cells were seeded in 96-well plates and infected with lung tissue supernatant in a dilution series (1–10$^{-4}$). After a 3-day incubation period, plates were fixed with 4% paraformaldehyde and permeabilized with 0.5% Triton X-100. Cells were stained with palivizumab (1:500) as primary antibody and horseradish peroxidase-labelled goat anti-human secondary antibody (1:500). Chloronaphtol (TFS, 34012) was added for 30 min at 37 °C.

## Graphs and statistical analyses

All graphs were prepared using GraphPad Prism 7 and 9 software. The same software was used for statistical analyses. Values of p ≤ 0.05 were considered as statistically significant. Data are represented as means ± standard error of the mean (SEM). Parasite burdens in the lungs were evaluated in 3–6 animals per time point. The flow cytometric data determining the various immune cell subsets in the lungs as well as the fold change calculated based on non-infected control mice, were derived from 3–5 animals per group and timepoint and were confirmed in at least one independent experiment. The FlexiVent™ study evaluating mechanical lung function parameters was accomplished with 4–5 animals per group. The co-infection experiments with RSV were performed with 3 mice per group and time point. All data were

analysed using a Kruskal-Wallis test (two-sided) with a Dunn's multiple comparisons test.

## Reporting summary

Further information on research design is available in the Nature Portfolio Reporting Summary linked to this article.

## Data availability

The authors declare that the data underlying the findings of this study are available within the paper and its Supplementary Information files and are available upon request. The source data underlying Figs. 1, 4–10 and Supplementary Figs. 7, 8 are provided as Source Data file. NCounter transcriptomics datasets have been made available at Gene Expression Omnibus (GEO accession GSE212622). Source data are provided with this paper.

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

## Acknowledgements

This work was supported by the Fonds Wetenschappelijk Onderzoek [www.fwo.be; grant numbers G033618N and G013118N, G.C.] and the University of Antwerp [www.uantwerpen.be; grant number TT-ZAPBOF 33049, G.C.]. The availability of tsetse flies was supported by the Slovak Research and Development Agency under the contract No. APVV-15-0604 entitled "Reduction of fecundity and trypanosomiasis control of tsetse flies by the application of sterile insect techniques and molecular methods", P.T. LMPH is a partner of the Excellence Centre 'Infla-Med' (www.uantwerpen.be/infla-med) and participates in COST Action CA21111.

## Author contributions

Design of research: D. Mabille and G.C. Performed experiments: D. Mabille, L.D., S.T., M.V., D. Montenye, M.G. and J.V.W. Provided tsetse flies: P.T. Analysed data: D. Mabille, L.D., S.T., M.V., D. Montenye, M.G., S.H., J.V.W., I.P., P.D., L.M., D.P.-M., J.P.T. and G.C. Wrote the manuscript: D. Mabille, L.D., S.T., M.V., D. Montenye, M.G., S.H., J.V.W., I.P., P.D., L.M., D.P.-M., J.P.T. and G.C.

## Competing interests

The authors declare no competing interests.
