## [Peer Review File · Nature Communications]

Impact of pulmonary African trypanosomes on the immunology and function of the lungREVIEWER COMMENTS

Reviewer #1 (Remarks to the Author):

In this work, Dorien Mabile and colleagues studied the impact of a naturally transmitted *T. brucei* infection in the lungs of mice. First, the authors quantified the parasite load in this and other organs and used scanning electron microscopy to identify the anatomical sites where parasites accumulate. Second, they described the immune cells recruited to the lungs during an infection (using histology, IFA and flow cytometry analysis). This analysis revealed an expected pattern of myeloid cell accumulation previously observed in *T. brucei*, but a surprising lack of recruitment of ($\alpha\beta$) T cells. Third, nanostring transcriptomic analysis revealed an immune signature overall compatible with the immune-phenotyping and consistent with literature, namely the presence of a Th1 proinflammatory response. Finally, the authors show that the infection does not "take the mouse's breath away".

The work is solid, the experiments are conclusive and most interpretations are correct. The authors use a diversity of tools that nicely complement each other. The paper is written clearly and most figures are self-explanatory and visually appealing (except for Fig. 8, see below). However, the thorough descriptive work was not followed by any functional study to test the importance of some of their observations. Besides, lung immunopathology was not assessed in a more terminal point of the infection.

MAJOR POINTS

The Immuno-phenotyping data and transcriptomics analysis reveals a wealth of data that could be explored functionally in innumerable directions.

- We found particularly intriguing that while immuno-phenotyping (Fig. 5) shows that $\alpha\beta$ T cells are not recruited to the lungs, transcriptomics analysis (Fig. 8) suggest $\gamma\delta$ T cells are recruited. This could be a very exciting observation on its own, but deserves more experiments: assess by flow cytometry the numbers of $\gamma\delta$ T cells in the lungs, confirm their activation by performing ex vivo cytokine staining and finally use knock-out or antibody depletion experiments to assess importance of $\gamma\delta$ T cells. One would expect that in the absence of $\gamma\delta$ T cells, parasite load might be higher with a distinct immunopathology and lung functional response (Fig. 10) could now be compromised.
- An alternative direction could be to test the susceptibility to secondary infections. This could be done by infecting mice via aerosol or intranasal route with at least one common pneumonia-inducing agent used in mice (e.g. *Klebsiella pneumoniae*, *Streptococcus pneumoniae*, Influenza A virus).

The authors elegantly show that lung function remains unaltered during infection. However, this raises the question of whether lung function would be affected if the infection was allowed to progress further, which should lead to higher lung immunopathology. In other words, it remains to be established if lung function is affected closer to natural death. If in this model day 21-PI is already close to natural death, or if later time-points cannot be assessed due to other experimental or ethical reasons this should be stated and discussed.

MINOR POINTS

1. While the title is very catchy, it would seem more appropriate for a review, not to a research paper. Besides, in the current version of this work, the answer to the question is "No", which is anticlimactic.
2. The description of the results should be followed by a short interpretation of their meaning. Currently, the results section describing the immune response lists the alterations without guiding the reader through the story. Given that not all readers of Nat Communications are immunologists, it may be helpful to concisely interpret the observations.
3. Line 73: "... twenty percent of HAT patients also suffer from respiratory symptoms that are

commonly attributed to secondary bacterial infections ...” The available data for respiratory symptoms in HAT comes from a study in T.b. rhodesiense infected patients. Of these, 20% had cough and 5.1% experienced both cough and dyspnoea. The original study by Kuepfer et al. (cited within ref [18]) attributed these 5.1% severe cases to cardiac insufficiency and make no mention of secondary bacterial infections. Accordingly, if the authors wish to make the link between T. brucei and secondary lung bacterial infections the appropriate original research should be cited.

4. Fig 1a. Authors should substitute the current figure (or add a new one) showing #parasites/mg tissue. This would clarify for instance whether the spleen represents roughly 80% of total parasite burden simply because overall there are very few parasites to be found by day 3 post-infection (and the very few that can be found are in the spleen) or whether this is not due to a low parasitemia bias and the spleen is actually a major early reservoir for the parasite.

5. Fig 1a. The authors should indicate which adipose tissue depot was analyzed.

6. The authors should explain the reason for choosing day 10 for most experiments. A parasitemia curve should also be shown as a supplementary figure.

7. Fig 1b. Data appears to have a bimodal distribution. Authors should provide a possible justification.

8. Fig 1d. Authors should label each panel; do they correspond to different days of infection?

9. Fig 2. Could authors show a schematics of the pulmonary anatomy, namely the sites where parasites were found: “lamina propria”, “the lung interstitium underneath the endothelium of capillaries, veins and arteries” and “the perichondrium of bronchial cartilage”

10. Line 112: Provide a small description of the morphological features used to identify this IC as myeloid.

11. Fig 4. Label three columns. Mouse 1,2,3. As immune cell infiltration is established in Figure 3 and quantitative data for neutrophils is provided in Figure 5, the authors could consider moving this figure to supplementary data.

12. Fig 5. The vast majority of total CD45+ cells reported in Fig 5a. are not accounted for by the populations reported in Fig 5c and 5e. Specifically, lungs from NI mice show 16-18 million CD45+ cells, however the sum of all subpopulations reported (which should account for the majority of lung ICs) accounts for only 20% of total CD45+ cells. This seems short, and it raises the question of which other ICs account for the remaining 80%. Authors should provide a possible justification..

13. Fig 5. The legend states that the data show 2 independent experiments with 3-5 mice per group. Some figures (e.g. 5a) clearly show more data points than that. Moreover, the use of overlapping empty circles makes it difficult to assess data distribution – consider changing to non-overlapping individual values.

14. T cell recruitment is apparently inexistent (Fig 5e-f). This is inconsistent with RNAseq showing presence of T cell-recruiting chemokines Cxcl9, Cxcl10, Ccl4, etc + Th1 response signature. Could this mean that T cell recruitment in this model is heavily $\gamma\delta$ -biased and this was missed because a TCR- β staining was used instead of CD3 for T cell identification?

15. Fig 5e-f. T cells should be labelled $\alpha\beta$ T cells.

16. Fig 6a. The conclusion is correct but the figure does not need to be so complex. Just show PCA C1/2.

17. Fig 7d. The panel “CD molecules” is redundant and not informative because there is no common function of those genes (other than the common “CD” name). This panel should be removed. Any non-redundant genes that the authors consider relevant could be included in the panel “Others”.

18. Fig 8. Panels are very confusing, they seem redundant and legend is very incomplete. What do colours represent? What do sizes of circles represent? What does "DeconRNAseq", "NNLM" and "COR" mean? Besides, the authors should describe in the Results what cellular deconvolution consists of.

19. Line 238 - If cellular deconvolution predicts the frequency of immune cell populations, the authors should show if these predictions correlate with the data obtained from flow cytometry analysis.

20. The authors discuss at length the possible implications of the transient eosinophilia observed by day 10 post-infection. However, eosinopenia has been previously shown to be a common feature of acute inflammation and bloodstream infections (i.e. Wibrow 2011; Bass 1980) – which is in agreement with the authors observations. This context should be provided in the Discussion.

Reviewer #2 (Remarks to the Author):

---The paper shows data to support the lung as a target organ, and the immunological role in the respiratory complications experienced by patients during African trypanosomiasis. The most interesting data presented is the significant reduction of B cells and downregulation of B cell receptor. The data also indicated no infection-associated pulmonary dysfunction, confirming the limited pulmonary clinical complications during the disease. There was also upregulation of interleukin 10, 2, 6, and Interferon gamma and alpha, in addition to negative immune checkpoint regulators.

---The paper addresses a very important aspect of tissue tropism in African trypanosomiasis and provide data to show the lung as a target of trypanosomes during sleeping sickness. African trypanosomes have historically been thought to be extracellular, however, several evidence shows the ability of the parasite to infect many other environments within the host. Interestingly, trypanosomes have been identified in tissues such as adipose tissues, testis spleen, heart muscle, brain.

---The data presented supports the conclusions in the paper.

---The methodologies are sound and are what is routinely used to answer similar questions in the field.

specific comments

Figure 1: Was the parasite burden between 7 – 21 dpi significant? What does the authors mean by "Data are represented as means \pm standard error of the mean of one independent repeat"? Is this an error?

Figure 3: Are the 2 sections from the same mice?

Figure 4: authors indicated that "Immunofluorescence staining showed the presence of neutrophils (Ly6G+ cells) with an indication of higher levels by 21 dpi ", however, there seems to be an increased no of neutrophils in the 10 dpi relative to 21 dpi. Why is this so? Was there any quantification of the signal? How was the "higher levels by 21 dpi" measured?

Figure 5: Are the statistical significance shown in c and e a comparison between the uninfected and either 10 dpi or 21 dpi? If that's the case, then this needs to be clarified.

--Authors need to improve on the quality of flow cytometry data provided in the in S7 and provide the gene description of the genes provided in the suppl excel document

Can authors explain or discuss what it will mean to have an upregulation of negative regulators of

T cell activation but have no differential regulation of T-cells as indicated in line 198-208. What is the implication of this on the pathogenesis of the disease?

One of the symptoms of sleeping sickness is the development of anemia. Authors indicated that "downregulation was observed at 21 dpi for Hamp encoding the antimicrobial protein hepcidin with a role in iron homeostasis" however authors were silent on what that means with respect the pathology induced in their experiments. I think this observation should be discussed.

Reviewer #3 (Remarks to the Author):

This study describes for the first time the presence of multiplying forms (long slender?) in the extravascular spaces surrounding the blood vessels of the alveoli and bronchi. The results further analyse the local immune response to infection by histopathological analysis, fluorescence microscopy, and flow cytometry. Together the data indicated an influx of monocytes, macrophages into the infected tissue and were consistent with the parasites inducing an inflammatory response. An interesting aspect of these studies was the reduction of eosinophils at the peak of the parasitaemia (10 dpi). The transcriptomic analysis of infected lung tissue indicated a strong IFN- γ and a Th1 proinflammatory response as well as an upregulation of negative immune checkpoint regulators; all which was consistent with a predominant M1 macrophage polarization typical of trypanosomiasis. However, the presence of trypanosomes in the lung did not appear to result in any infection-associated pulmonary dysfunction using standard tests of such function.

Overall this is an interesting study. There is a direct demonstration of trypanosomes in the interstitial spaces of the lung, a solid immunological analysis of the infected tissue and a comparative transcriptomic analysis of infected v non-infected tissue; this is a reasonable body of work. There is no real issue with the results per se, although it could be argued that some of findings are perhaps overstated. This main issue is whether the impact of the findings are of high significance and/or of wide appeal.

In this regard there are two potentially interesting outcomes from the study but there are limitations

First, the suggestion by the authors that the substantial reduction of eosinophils, B cells and NK cells in trypanosome infected lung tissue may render individuals more susceptible to opportunistic infections of the lung.

The authors should present relevant data to support this claim, for example by demonstrating a greater susceptibility to a viral (COVID19, flu) or bacterial (MTb, pertussis, strep) respiratory infection in lung-infected mice. Such a demonstration would greatly improve the impact and significance of their findings. However as it stands this idea is speculative and also may be at odds with the lack of infection-associated pulmonary dysfunction.

Second, while there is a clear demonstration of trypanosomes in the extravascular spaces in the lungs, i.e. the excellent SEM data (Fig. 2), it is unclear if this lung population has any special significance within the parasite life cycle. Trypanosomes are exclusively extracellular parasites but it has been long recognized they are not restricted to the vasculature or CNS but are found in many interstitial spaces. It is highly likely that trypanosomes, being facultative anaerobes, can penetrate the entire tissue space of the host.

The question is whether there is a specific function associate with forms in these compartments. There is some evidence that metabolically adapted forms, distinct from the proliferative long slender form, may colonise the adipose tissue and experimental infections have shown that the skin can harbour populations of trypanosomes that can be transmitted to the tsetse vector, even in the absence of detectable parasitaemia, and may perhaps have diagnostic potential.

The data here suggest that the lung population is ~100-500 trypanosomes/mg tissue and interestingly remains relatively, even at the peak of the parasitaemia. So, compared to the parasite burden within the vasculature this is a relatively small population and given its location cannot be directly involved in transmission during tsetse fly feeding.

The authors suggest that the high oxygen tension may affect metabolic characteristics of the pulmonary parasites. While this possibility is interesting, it was not explored. It might be expected

that high oxygen tensions would suit well the glycolytic metabolism of proliferating long slender forms, allowing them to re-oxidise glycolytic NADH via the TAO. Again if it could be demonstrated that these parasites has specific features and functions that made them distinct from the proliferative long slender form then the impact of the study would increase.

The other interesting aspect was the observation of secretion of extracellular vesicles (Fig. 2 g & h). Although extracellular vesicles and parasite to parasite communication have been reported in vitro (ref 19) the images in Fig. 2 are first in vivo examples. In terms of size and location they resemble similar structures observed when trypanosomes were treated with nanobody antibodies against VSG (Hempelmann A. et al., 2021, Cell Reports 37, 109923), potentially a cell stress response?

In the discussion the authors say that pulmonary parasites exhibited extensive cell-cell communication through the release of EVs. This would appear to over-state the case. There are two trypanosomes that exhibit these EVs (Fig. 2g and again in Fig. 2h). However these EVs do not appear to be visible in any of the other images there is no evidence to support the claim of cell to cell communication except proximity. The authors are entitled to suggest this possibility but not to claim it as a fact.

Two minor points;

The authors should include the parasitaemia data in Fig. 1c, according to the methods (p28) they have this data

It is not clear from the legend (Fig. 1) or the methods when the bioluminescent imaging (BLI) was performed other than to say it was following an infective bite

I found this to be an interesting study and was impressed by the quality of the imaging in Fig. 2. However, the results are primarily descriptive rather than demonstrative in terms of impact. The authors need top consider the following questions to address this impact issue.

Do the presence of trypanosomes in the extravascular spaces of the lung render the host more susceptible to opportunistic infections of the lung?

Have these parasites a demonstrable special role or function in the life cycle?

Answer letter

To address the received comments from the reviewers, we have performed additional experimental work. We have integrated these new data and have modified the manuscript to address all major and minor comments raised. Please find below a point-by-point answer to the questions. We have uploaded a revised version with changes tracked.

REVIEWER COMMENTS

Reviewer #1 (Remarks to the Author):

In this work, Dorien Mabile and colleagues studied the impact of a naturally transmitted *T. brucei* infection in the lungs of mice. First, the authors quantified the parasite load in this and other organs and used scanning electron microscopy to identify the anatomical sites where parasites accumulate. Second, they described the immune cells recruited to the lungs during an infection (using histology, IFA and flow cytometry analysis). This analysis revealed an expected pattern of myeloid cell accumulation previously observed in *T. brucei*, but a surprising lack of recruitment of ($\alpha\beta$) T cells. Third, nanostring transcriptomic analysis revealed an immune signature overall compatible with the immune-phenotyping and consistent with literature, namely the presence of a Th1 proinflammatory response. Finally, the authors show that the infection does not “take the mouse’s breath away”.

The work is solid, the experiments are conclusive and most interpretations are correct. The authors use a diversity of tools that nicely complement each other. The paper is written clearly and most figures are self-explanatory and visually appealing (except for Fig. 8, see below). However, the thorough descriptive work was not followed by any functional study to test the importance of some of their observations. Besides, lung immunopathology was not assessed in a more terminal point of the infection.

MAJOR POINTS

The immuno-phenotyping data and transcriptomics analysis reveals a wealth of data that could be explored functionally in innumerable directions.

- We found particularly intriguing that while immuno-phenotyping (Fig. 5) shows that $\alpha\beta$ T cells are not recruited to the lungs, transcriptomics analysis (Fig. 8) suggest $\gamma\delta$ T cells are recruited. This could be a very exciting observation on its own, but deserves more experiments: assess by flow cytometry the numbers of $\gamma\delta$ T cells in the lungs, confirm their activation by performing ex vivo cytokine staining and finally use knock-out or antibody depletion experiments to assess importance of $\gamma\delta$ T cells. One would expect that in the absence of $\gamma\delta$ T cells, parasite load might be higher with a distinct immunopathology and lung functional response (Fig. 10) could now be compromised.

We agree that this finding is intriguing and of potential importance for regulating parasitemia levels and the development of immunopathology. We performed an additional infection experiment using a novel flow cytometry staining panel to identify $\gamma\delta$ T cells. A significant increase in the number of $\gamma\delta$ T cells was detected in the lungs upon infection with African trypanosomes, corroborating the cellular deconvolution data (see figure 4e-f). Due to time constraints we were not able to introduce the

appropriate knockout mouse model and obtain ethical clearance to scrutinize the role of these cells during infection. We have now included the new data to the manuscript and also briefly highlighted this observation in the abstract and discussion.

- An alternative direction could be to test the susceptibility to secondary infections. This could be done by infecting mice via aerosol or intranasal route with at least one common pneumonia-inducing agent used in mice (e.g. *Klebsiella pneumoniae*, *Streptococcus pneumoniae*, Influenza A virus). Upon suggestion by this reviewer, we tested the susceptibility of *T. brucei* infected mice to intranasal infection with Respiratory Syncytial Virus (RSV). Briefly, six mice were infected with trypanosomes via the bite of an infected tsetse fly. The other 6 mice were not infected with trypanosomes. At day 13 post trypanosome infection, all mice were infected via intranasal administration with RSV. Autopsy was performed at day 4 and 8 post RSV infection. Lung tissue was collected for RNA extraction and also homogenized to collect the virus. RT-qPCR and a plaque assay were performed to determine the viral load in the lungs. Despite the small group size, a clear and significant trend was visible: the non-infected control group showed an expected peak of viral load at day 4 post infection with a reduction by day 8 (and absence of viable virus following the plaque assay at day 8). The trypanosome-infected group on the other hand started off with a lower viral load, likely due to prior immune activation, but mice were not able to control the virus by day 8.

Given the recorded impact on the RSV infection kinetics, we decided to keep the statement in the manuscript that African trypanosomes may affect pulmonary susceptibility to other (opportunistic) infections. To support this statement, we have included the data in the revised manuscript and included a new figure (Fig. 10).

The authors elegantly show that lung function remains unaltered during infection. However, this raises the question of whether lung function would be affected if the infection was allowed to progress further, which should lead to higher lung immunopathology. In other words, it remains to be established if lung function is affected closer to natural death. If in this model day 21-PI is already close to natural death, or if later time-points cannot be assessed due to other experimental or ethical reasons this should be stated and discussed.

In this model, mice can survive up to approximately 40 dpi. However, a large proportion of mice already succumb following the first peak of parasitemia (around 10 dpi) and around 45 percent by 21 dpi. We added parasitemia and survival data for our tsetse transmitted *T. brucei* model (supplementary figure S8). We chose 21 dpi as a representative time-point of late-stage infection with a CNS involvement as observed in stage II HAT patients.

MINOR POINTS

1. While the title is very catchy, it would seem more appropriate for a review, not to a research paper. Besides, in the current version of this work, the answer to the question is “No”, which is anticlimactic.

The title was changed to: Identification of a pulmonary parasite niche for African trypanosomes: parasitological observations, immunological correlates and effects on lung function

2. The description of the results should be followed by a short interpretation of their meaning. Currently, the results section describing the immune response lists the alterations without guiding the

reader through the story. Given that not all readers of Nat Communications are immunologists, it may be helpful to concisely interpret the observations.

Given the restriction in word count we opted to concisely describe the results and provide a more extensive interpretation in the Discussion section.

3. Line 73: "... twenty percent of HAT patients also suffer from respiratory symptoms that are commonly attributed to secondary bacterial infections ...". The available data for respiratory symptoms in HAT comes from a study in *T.b. rhodesiense* infected patients. Of these, 20% had cough and 5.1% experienced both cough and dyspnoea. The original study by Kuepfer et al. (cited within ref [18]) attributed these 5.1% severe cases to cardiac insufficiency and make no mention of secondary bacterial infections. Accordingly, if the authors wish to make the link between *T. brucei* and secondary lung bacterial infections the appropriate original research should be cited.

We would like to thank the reviewer for correctly pointing this out. We have modified the text accordingly.

Line 76-81: An underreported aspect is that a substantial proportion of HAT patients also suffers from respiratory symptoms. In Tanzania and Uganda more than twenty percent of stage-II patients were reported with a cough (20%) or dyspnoea (7%)^{18,19}. Although the majority of severe dyspnoea cases were commonly attributed to cardiac insufficiency, the role of secondary viral/bacterial/fungal infections remains largely unknown. Bacterial bronchopneumonia is one of the complications that has been reported in animal trypanosomiasis^{20,21}.

4. Fig 1a. Authors should substitute the current figure (or add a new one) showing #parasites/mg tissue. This would clarify for instance whether the spleen represents roughly 80% of total parasite burden simply because overall there are very few parasites to be found by day 3 post-infection (and the very few that can be found are in the spleen) or whether this is not due to a low parasitemia bias and the spleen is actually a major early reservoir for the parasite.

A new panel has been added to figure 1 (Fig. 1b) to show the number of parasites per mg tissue.

5. Fig 1a. The authors should indicate which adipose tissue depot was analyzed.

Gonadal adipose tissue. This information has been added to the figure legend and text.

Line 92-93: On the other hand, a steep increase in burden was observed in the sampled gonadal adipose tissue, as described previously¹³.

Legend Fig. 1: Parasite burden in the lungs, heart, spleen, kidney, brain and gonadal adipose tissue of naturally infected mice.

6. The authors should explain the reason for choosing day 10 for most experiments. A parasitemia curve should also be shown as a supplementary figure.

A supplementary figure (Fig. S8) has been added to provide parasitemia and survival data of our naturally transmitted *T. brucei* model. Experiments were performed at day 10, a timepoint at which the majority of mice exhibit an initial peak parasitemia. From an ethical viewpoint, we prefer the

infection experiment to be conducted within a time frame of 14 days after infection, as already several mice succumb from infection. For some critical experiments we also included 21 dpi, representing a late stage infection with CNS involvement.

7. Fig 1b. Data appears to have a bimodal distribution. Authors should provide a possible justification.

This can be explained by the lower parasite burden in one of the animals. The burdens in this mouse were consistently lower in all measured lung sections.

8. Fig 1d. Authors should label each panel; do they correspond to different days of infection?

The different panels correspond to different mice all measured at 7 dpi. This information has been added to the figure legend.

Legend Fig.1: Organ distribution of *T. b. brucei*^{Ppy^{REG}} parasites determined via bioluminescent imaging at 7 dpi in three different mice.

9. Fig 2. Could authors show a schematics of the pulmonary anatomy, namely the sites where parasites were found: “lamina propria”, “the lung interstitium underneath the endothelium of capillaries, veins and arteries” and “the perichondrium of bronchial cartilage”

An additional schematic figure (Fig. 11) has been added to show the histological features of bronchus, bronchiolus and alveolus and the corresponding parasite localization.

10. Line 112: Provide a small description of the morphological features used to identify this IC as myeloid.

We believe that this is a macrophage based on the plate like appearance and ring like membrane surface with several protrusions of varying length. A lymphocyte would probably be more spherical. However, we cannot be 100% sure and have therefore modified the figure and corresponding legend to “white blood cell”.

Line 110-111: White blood cells were observed in close vicinity of the multiplying trypanosomes (Fig. 2f).

11. Fig 4. Label three columns. Mouse 1,2,3. As immune cell infiltration is established in Figure 3 and quantitative data for neutrophils is provided in Figure 5, the authors could consider moving this figure to supplementary data.

We agree with the reviewer and have moved this figure to the supplementary data. Column titles have been added to the figure. The figure numbering and referencing in the text has been updated.

12. Fig 5. The vast majority of total CD45+ cells reported in Fig 5a. are not accounted for by the populations reported in Fig 5c and 5e. Specifically, lungs from NI mice show 16-18 million CD45+ cells, however the sum of all subpopulations reported (which should account for the majority of lung ICs) accounts for only 20% of total CD45+ cells. This seems short, and it raises the question of which other ICs account for the remaining 80%. Authors should provide a possible justification.

We acknowledge the reviewer for pointing this out. The calculation of the total number of CD45+ cells was updated. The figure was corrected accordingly.

13. Fig 5. The legend states that the data show 2 independent experiments with 3-5 mice per group. Some figures (e.g. 5a) clearly show more data points than that. Moreover, the use of overlapping empty circles makes it difficult to assess data distribution – consider changing to non-overlapping individual values.

This statement has been adjusted to “at least” 2 independent experiments since data have been added and experiments were repeated for this rebuttal. Figures containing non-overlapping values was not possible due to the high number of replicates. The graphs, however, were rearranged to allow a slight enlargement of the different panels. In addition, we have now included all raw data in a supplemental Excel-file.

14. T cell recruitment is apparently inexistent (Fig 5e-f). This is inconsistent with RNAseq showing presence of T cell-recruiting chemokines Cxcl9, Cxcl10, Ccl4, etc + Th1 response signature. Could this mean that T cell recruitment in this model is heavily $\gamma\delta$ -biased and this was missed because a TCR- β staining was used instead of CD3 for T cell identification?

There is a significant increase of CD4⁺/CD25⁺ $\alpha\beta$ T cells, compatible with recruitment and activation of T cells. We included an additional flow cytometry panel for the identification of CD3⁺ $\gamma\delta$ T cell, demonstrating a significant increase in $\gamma\delta$ T cells upon infection (Figure 4e-f).

15. Fig 5e-f. T cells should be labelled $\alpha\beta$ T cells.

This has been adjusted in the corresponding figure.

16. Fig 6a. The conclusion is correct but the figure does not need to be so complex. Just show PCA C1/2.

This has been adjusted in the corresponding figure.

17. Fig 7d. The panel “CD molecules” is redundant and not informative because there is no common function of those genes (other than the common “CD” name). This panel should be removed. Any non-redundant genes that the authors consider relevant could be included in the panel “Others”.

This has been adjusted following the reviewer’s recommendation.

18. Fig 8. Panels are very confusing, they seem redundant and legend is very incomplete. What do colours represent? What do sizes of circles represent? What does “DeconRNAseq”, “NNLM” and “COR” mean? Besides, the authors should describe in the Results what cellular deconvolution consists of.

An explanation of cellular deconvolution has been added to the text and the methods section. Additional information has been added to the figure legend for clarification. Multiple computational deconvolution methods were used within the OmicsPlayground software that use different statistical frameworks, revealing recurrent cellular signatures which strengthens (in addition to the flow cytometry data) the immunological changes documented in the manuscript.

Line 194-195: Cellular deconvolution analysis, which estimates the proportion of cells in a tissue based on the gene expression profiles,

Line 199-201: These cellular fingerprints were conserved over the different applied computational methods for cellular deconvolution (Fig. 7b) and supported by the flow cytometry data (Fig. 4).

Line 400-402: Cellular deconvolution of the transcript data was based on various computational algorithms that use a variety of statistical frameworks (DeconRNAseq, DCQ, I-NNLS, NNLM, NNLM.lin, NNLM.rnk, cor, meta, meta.prod).

Legend fig. 7: Gene ontology analysis and cell type mapping of *T. b. brucei* infected lung tissue. **a** Activation matrix visualizing the enrichment of GO terms across the different comparisons (21 dpi vs 10 dpi, 21 dpi vs NI, 10 dpi vs NI). Red colour corresponds to upregulation and blue colour corresponds to downregulation of the corresponding pathway. Circle size corresponds to the number of genes involved in the specific pathway. **b** Heatmap visualizing the distribution of inferred immune cell types based on nCounter digital transcriptomics for different computational deconvolution methods (DeconRNAseq, DCQ, I-NNLS, NNLM, NNLM.lin, NNLM.rnk, cor, meta, meta.prod) in the OmicsPlayground software. **c** Overview dot plot of the cell type mapping. Some major changes are highlighted in yellow. Circle size corresponds to the number of genes mapped to the corresponding cell type and the colour represent the q-value (darker colour is lower q-value).

19. Line 238 - If cellular deconvolution predicts the frequency of immune cell populations, the authors should show if these predictions correlate with the data obtained from flow cytometry analysis.

The cellular deconvolution provides an indirect quantification of immune cell populations. Nevertheless, the cellular composition determined by flow cytometry support the deconvolution data (increase in macrophages, $\gamma\delta$ T cells, T cell activation, and a decrease in B cells and eosinophils).

20. The authors discuss at length the possible implications of the transient eosinophilia observed by day 10 post-infection. However, eosinopenia has been previously shown to be a common feature of acute inflammation and bloodstream infections (i.e. Wibrow 2011; Bass 1980) – which is in agreement with the authors observations. This context should be provided in the Discussion.

This information and observations of eosinopenia in COVID-19 patients has been added to the discussion section.

Line 286-290: In addition, eosinopenia has been previously shown to be a common feature of acute inflammation and bloodstream infections^{55,56}. Also in COVID-19 patients eosinopenia is frequently observed and confers a higher risk for developing severe disease and for requiring admission to the intensive care unit. Whether eosinophils play a major role during a *T. brucei* infection is still unknown⁵⁷.

Reviewer #2 (Remarks to the Author):

---The paper shows data to support the lung as a target organ, and the immunological role in the respiratory complications experienced by patients during African trypanosomiasis. The most interesting data presented is the significant reduction of B cells and downregulation of B cell receptor. The data also indicated no infection-associated pulmonary dysfunction, confirming the limited

pulmonary clinical complications during the disease. There was also upregulation of interleukin 10, 2, 6, and Interferon gamma and alpha, in addition to negative immune checkpoint regulators.

---The paper addresses a very important aspect of tissue tropism in African trypanosomiasis and provide data to show the lung as a target of trypanosomes during sleeping sickness. African trypanosomes have historically been thought to be extracellular, however, several evidence shows the ability of the parasite to infect many other environments within the host. Interestingly, trypanosomes have been identified in tissues such as adipose tissues, testis spleen, heart muscle, brain.

---The data presented supports the conclusions in the paper.

---The methodologies are sound and are what is routinely used to answer similar questions in the field.

specific comments

Figure 1: Was the parasite burden between 7 – 21 dpi significant? What does the authors mean by “Data are represented as means \pm standard error of the mean of one independent repeat”? Is this an error?

The data is the result of one *in vivo* experiment with at least 3 mice per group. The statistical analysis was added to Figure 1c.

Figure 3: Are the 2 sections from the same mice?

The two sections shown for 21 dpi are from different mice. For the other timepoints only 1 representative image is shown since no histopathology was observed. A statement has been added to the figure legend to specify that the observations were made in 3 different animals per time point.

Figure 3 legend: Similar observations were made in 3 mice per group and representative images are shown.

Figure 4: authors indicated that “Immunofluorescence staining showed the presence of neutrophils (Ly6G⁺ cells) with an indication of higher levels by 21 dpi “, however, there seems to be an increased no of neutrophils in the 10 dpi relative to 21 dpi. Why is this so? Was there any quantification of the signal? How was the “higher levels by 21 dpi” measured?

Quantification of the immunofluorescence did not yield significant differences between the different timepoints. The quantitative flow cytometric data however clearly showed an increase in neutrophils at 21 dpi. The figure has been moved to the supplementary information and the statement in the text has been modified accordingly.

Line 119-120: Immunofluorescence staining showed the presence of neutrophils (Ly6G⁺ cells) during infection (Supplementary fig. S4).

Figure 5: Are the statistical significance shown in c and e a comparison between the uninfected and either 10 dpi or 21 dpi? If that’s the case, then this needs to be clarified.

This is correct. A clarification has been added to the figure legend.

Legend Fig. 5: Statistical comparisons were made using the Kruskal-Wallis test between non-infected and either 10 dpi or 21 dpi.

--Authors need to improve on the quality of flow cytometry data provided in the in S7 and provide the gene description of the genes provided in the suppl excel document

Figure S7 has been updated and a supplementary excel file containing the gene descriptions has been provided. In addition, expression data have been uploaded to the Gene Expression Omnibus (GEO accession GSE212622; <https://www.ncbi.nlm.nih.gov/geo/query/acc.cgi?acc=GSE212622>).

Can authors explain or discuss what it will mean to have an upregulation of negative regulators of T cell activation but have no differential regulation of T-cells as indicated in line 198-208. What is the implication of this on the pathogenesis of the disease?

The negative checkpoint inhibitors will inhibit T cell proliferation. We detected increased levels of activated CD4⁺/CD25⁺ αβ T cells in the lungs and the checkpoint inhibitors are expected to keep further T cell activation and expansion in check. Recent observations indeed show that CXCR6⁺ CD4⁺ T cells can play a detrimental role, as they are responsible for liver tissue injury and early mortality. The role of this observation in the development of lung pathology is unclear. Additional experiments performed in response to the reviewers have demonstrated a significant increase of γδ T cells upon infection. These cells have been previously described to play a role in disease control in trypanotolerant animals. These aspects have now been discussed more in the manuscript. The discussion has been extended with inclusion of appropriate references.

Lines 297-308: Cellular deconvolution provides very strong support (p-values < 10⁻⁴ to <10⁻⁶) for the induction of γδ T cells, a cell type known to regulate bacterial clearance in infected lungs via the production of IL-17 and neutrophil recruitment and to protect against lung fibrosis through IL-22⁶⁰. In our lung transcriptome data, *Il17* and *Il22* were not found to be upregulated upon trypanosome infection. Cellular immunoprofiling confirmed a significant increase of γδ T cell numbers alongside CD4⁺ CD25⁺ αβ T cells. As this increase of CD4⁺ CD25⁺ αβ T cells was not accompanied by *Foxp3* upregulation, this indicates the elevated presence of activated T cells rather than regulatory T cells. The significant upregulation of several immune checkpoint genes may contribute to keeping T cell activation and proliferation in check and to T cell suppression, a hallmark of trypanosomiasis. This regulation may be important to control inflammation and tissue injury, as accumulation of CXCR6⁺ CD4⁺ T lymphocytes is pivotal in inducing liver pathology and early mortality. The role in lung pathology remains to be understood.

One of the symptoms of sleeping sickness is the development of anemia. Authors indicated that “downregulation was observed at 21 dpi for *Hamp* encoding the antimicrobial protein hepcidin with a role in iron homeostasis” however authors were silent on what that means with respect the pathology induced in their experiments. I think this observation should be discussed.

Hamp plays an important role in iron metabolism and erythropoiesis. We can only speculate what the role of hepcidin would be in the lung. *Hamp*^{-/-} mice do not show an altered parasitemia, so hepcidin does not seem to affect trypanosomes directly. However, as an antimicrobial peptide with antibacterial and antifungal activity, downregulation may affect susceptibility of the lung to opportunistic infections. Through additional co-infection experiments, we have broadened the scope of the manuscript by

illustrating that trypanosome infection and the immunological changes in the lung can affect susceptibility to opportunistic infections. We have modified the manuscript to include these new data and to discuss the putative role of Hamp.

Reviewer #3 (Remarks to the Author):

This study describes for the first time the presence of multiplying forms (long slender?) in the extravascular spaces surrounding the blood vessels of the alveoli and bronchi. The results further analyse the local immune response to infection by histopathological analysis, fluorescence microscopy, and flow cytometry. Together the data indicated an influx of monocytes, macrophages into the infected tissue and were consistent with the parasites inducing an inflammatory response. An interesting aspect of these studies was the reduction of eosinophils at the peak of the parasitaemia (10 dpi). The transcriptomic analysis of infected lung tissue indicated a strong IFN- γ and a Th1 proinflammatory response as well as an upregulation of negative immune checkpoint regulators; all which was consistent with a predominant M1 macrophage polarization typical of trypanosomiasis. However, the presence of trypanosomes in the lung did not appear to result in any infection-associated pulmonary dysfunction using standard tests of such function.

Overall, this is an interesting study. There is a direct demonstration of trypanosomes in the interstitial spaces of the lung, a solid immunological analysis of the infected tissue and a comparative transcriptomic analysis of infected v non-infected tissue; this is a reasonable body of work. There is no real issue with the results per se, although it could be argued that some of findings are perhaps overstated. This main issue is whether the impact of the findings are of high significance and/or of wide appeal.

In this regard there are two potentially interesting outcomes from the study but there are limitations. First, the suggestion by the authors that the substantial reduction of eosinophils, B cells and NK cells in trypanosome infected lung tissue may render individuals more susceptible to opportunistic infections of the lung. The authors should present relevant data to support this claim, for example by demonstrating a greater susceptibility to a viral (COVID19, flu) or bacterial (MTb, pertussis, strep) respiratory infection in lung-infected mice. Such a demonstration would greatly improve the impact and significance of their findings. However, as it stands this idea is speculative and also may be at odds with the lack of infection-associated pulmonary dysfunction.

As also suggested by reviewer 1, we tested the susceptibility of *T. brucei* infected mice to intranasal infection with Respiratory Syncytial Virus (RSV). Six mice were infected with trypanosomes via the bite of an infected tsetse fly. The other 6 mice were not infected with trypanosomes. At day 13 post trypanosome infection, all mice were infected via intranasal administration with RSV. Autopsy was performed at day 4 and 8 post RSV infection. Lung tissue was collected for RNA extraction and also homogenized to collect the virus. RT-qPCR and a plaque assay were performed to determine the viral load in the lungs.

Despite the rather small group size, a clear trend was visible: the non-infected control group showed a peak of viral load at day 4 post infection with a reduction in viral burdens at day 8 (and absence of viable virus following the plaque assay at day 8). The trypanosome-infected group on the other hand started off with a lower viral load likely due the already activated immune system but were not able to control the virus by day 8 post RSV infection. Additionally, we determined the bacterial burdens in

the lungs to gain insight in of trypanosome-infected mice and non-infected control mice via RT-qPCR targeting bacterial 16S rRNA. These new data are included and discussed in the revised manuscript.

Second, while there is a clear demonstration of trypanosomes in the extravascular spaces in the lungs, i.e. the excellent SEM data (Fig. 2), it is unclear if this lung population has any special significance within the parasite life cycle. Trypanosomes are exclusively extracellular parasites but it has been long recognized they are not restricted to the vasculature or CNS but are found in many interstitial spaces. It is highly likely that trypanosomes, being facultative anaerobes, can penetrate the entire tissue space of the host.

The question is whether there is a specific function associated with forms in these compartments. There is some evidence that metabolically adapted forms, distinct from the proliferative long slender form, may colonise the adipose tissue and experimental infections have shown that the skin can harbour populations of trypanosomes that can be transmitted to the tsetse vector, even in the absence of detectable parasitaemia, and may perhaps have diagnostic potential.

The data here suggest that the lung population is ~100-500 trypanosomes/mg tissue and interestingly remains relatively, even at the peak of the parasitaemia. So, compared to the parasite burden within the vasculature this is a relatively small population and given its location cannot be directly involved in transmission during tsetse fly feeding.

The authors suggest that the high oxygen tension may affect metabolic characteristics of the pulmonary parasites. While this possibility is interesting, it was not explored. It might be expected that high oxygen tensions would suit well the glycolytic metabolism of proliferating long slender forms, allowing them to re-oxidise glycolytic NADH via the TAO. Again if it could be demonstrated that these parasites have specific features and functions that made them distinct from the proliferative long slender form then the impact of the study would increase.

This is a very valid point given the interesting changes in metabolic characteristics of parasites collected from adipose tissue as elegantly shown by the Figueiredo group. We are currently planning an *in vivo* experiment to isolate DsRED+ parasites for differential transcriptomic profiling from different tissues (blood, spleen, skin and lungs) which will be the topic of a follow-up publication. We believe that the lung trypanosomes will exhibit a different profile than those in the other tissues.

The other interesting aspect was the observation of secretion of extracellular vesicles (Fig. 2 g & h). Although extracellular vesicles and parasite to parasite communication have been reported *in vitro* (ref 19) the images in Fig. 2 are first *in vivo* examples. In terms of size and location they resemble similar structures observed when trypanosomes were treated with nanobody antibodies against VSG (Hempelmann A. et al., 2021, Cell Reports 37, 109923), potentially a cell stress response? In the discussion the authors say that pulmonary parasites exhibited extensive cell-cell communication through the release of EVs. This would appear to over-state the case. There are two trypanosomes that exhibit these EVs (Fig. 2g and again in Fig. 2h). However, these EVs do not appear to be visible in any of the other images there is no evidence to support the claim of cell to cell communication except proximity. The authors are entitled to suggest this possibility but not to claim it as a fact.

As suggested by the reviewer the statements about the extent of secretion and the role in cell-to-cell communication were moderated. However, although only 3 images of EVs are shown in the manuscript (Fig 2g, 2h, S3), the bulk of SEM images which we acquired contained multiple examples of EV secretion by the parasite. This is a unique observation that we really feel is biologically relevant. We performed

similar studies in the skin (Caljon et, 2019) and did not observe the “bead on a string” EV release. We have modified the manuscript, with inclusion of the additional reference (Hempelmann et al. , 2021).

Line 256-259: Pulmonary parasites cluster together in nests and seemed to exhibit cell-cell communication through the release of EVs²² which, to our knowledge, has not yet been documented in an *in vivo* setting. Similar ultrastructural studies in infected skin could not reveal the relatively frequent EV release as “beads on a string” observed in the lungs⁴⁷ and *in vitro*^{22,48}.
Two minor points;

The authors should include the parasitaemia data in Fig. 1c, according to the methods (p28) they have this data

A parasitemia and survival curve is now included in the supplementary information (Figure S8).

It is not clear from the legend (Fig. 1) or the methods when the bioluminescent imaging (BLI) was performed other than to say it was following an infective bite

BLI was performed at 7 dpi. This information has been added to the figure legend.

Legend Fig. 1: Organ distribution of *T. b. brucei* AnTat1.1E^{PpyRE9} parasites determined via bioluminescent imaging at 7 dpi in three different mice.

I found this to be an interesting study and was impressed by the quality of the imaging in Fig. 2. However, the results are primarily descriptive rather than demonstrative in terms of impact. The authors need to consider the following questions to address this impact issue. Do the presence of trypanosomes in the extravascular spaces of the lung render the host more susceptible to opportunistic infections of the lung? Have these parasites a demonstrable special role or function in the life cycle?

In response to this comment, an RSV-trypanosome co-infection experiment was performed. This study strongly pointed in the direction of an increased susceptibility of trypanosome-infected animals to pulmonary infections. We have added this new information to the revised version of the manuscript. Related to the life cycle, we are uncertain whether the lung parasites undergo specific metabolic changes or a differentiation that would be of importance for systemic infection or transmission. We will undertake differential transcriptomic profiling of parasites from different tissues (blood, spleen, skin and lungs) for further characterization.

REVIEWERS' COMMENTS

Reviewer #1 (Remarks to the Author):

The authors have carefully considered my suggestions and addressed my most important concerns. Figure 11 is beautiful!

Reviewer #2 (Remarks to the Author):

I am happy with the revisions, and only have very minor suggestions I would ask the authors to consider before the manuscript can be accepted for publication.

Figure 10 - authors can change "Impact of T. brucei on co-infections" To "Impact of T. b. brucei on co-infections"

Figure 11 – Authors can differentiate between layers eg like EP { and BM { etc so that a reader can different between the different parts

Reviewer #3 (Remarks to the Author):

First, did presence of trypanosomes in the extravascular spaces of the lung have an effect on viral/bacterial infections of the lung and second were these lung resident forms a new tissue specific form in the host.

The authors have addressed the first issue of whether the presence of trypanosomes in the extravascular spaces of the lung render the host more susceptible to opportunistic infections of the lung. The results the RSV study suggest that they do, and I think this experiment certainly adds to the paper.

However, It is not clear at present if these lung resident forms are a new tissue specific form, and this is acknowledged by the authors and in fairness the authors do not claim this is the case.

On balance I feel that the impact of the work has now greater, and publication is justified subject to one qualification.

My only remaining concern is the emphasis on the apparent extracellular vesicles and their potential role in cell- to-cell communication in Fig 2. The authors have moderated their reference to ECVs and possible cell to cell communication:

They say:

Pulmonary parasites cluster together in nests and seemed to exhibit cell-cell communication through the release of EVs.

I think this statement is still an over interpretation. While these interesting structures are associated with the trypanosomes, they are not associated with all trypanosomes in the image, and it is not 100% certain that these structures are of trypanosomal origin. They probably are but this has not been demonstrated. The statement also implies that these structures have functional roles, which they may, but equally we cannot be certain that they do not represent an aberrant morphological feature, perhaps a stress response, in the population.

I think the authors need to be more equivocal here, the images present are interesting and novel and but at present the significance and role of these structures are uncertain. A clearer statement to this effect should be present in the discussion.

A last minor point is line 241

Our present research has identified the lungs as a prominent site of parasite proliferation during natural

The data and the relative parasite load do not justify the term prominent, the authors have shown that the lungs are a site of infection but clearly the parasite load is the same as in the vasculature.

REVIEWERS' COMMENTS

We would like to thank the reviewers for the very constructive comments. This has greatly increased the quality of the paper.

Reviewer #1 (Remarks to the Author):

The authors have carefully considered my suggestions and addressed my most important concerns. Figure 11 is beautiful!

Reviewer #2 (Remarks to the Author):

I am happy with the revisions, and only have very minor suggestions I would ask the authors to consider before the manuscript can be accepted for publication.

Figure 10 - authors can change "Impact of T. brucei on co-infections" To "Impact of T. b. brucei on co-infections"

This has been adjusted in the figure legend.

Figure 11 – Authors can differentiate between layers eg like EP { and BM { etc so that a reader can differentiate between the different parts

This was adjusted in the corresponding figure.

Reviewer #3 (Remarks to the Author):

First, did presence of trypanosomes in the extravascular spaces of the lung have an effect on viral/bacterial infections of the lung and second were these lung resident forms a new tissue specific form in the host.

The authors have addressed the first issue of whether the presence of trypanosomes in the extravascular spaces of the lung render the host more susceptible to opportunistic infections of the lung. The results the RSV study suggest that they do, and I think this experiment certainly adds to the paper.

However, It is not clear at present if these lung resident forms are a new tissue specific form, and this is acknowledged by the authors and in fairness the authors do not claim this is the case.

On balance I feel that the impact of the work has now greater, and publication is justified subject to one qualification.

My only remaining concern is the emphasis on the apparent extracellular vesicles and their potential role in cell- to-cell communication in Fig 2. The authors have moderated their reference to ECVs and possible cell to cell communication:

They say:

Pulmonary parasites cluster together in nests and seemed to exhibit cell-cell communication through the release of EVs. I think this statement is still an over interpretation. While these interesting structures are associated with the trypanosomes, they are not associated with all trypanosomes in the image, and it is not 100% certain that these structures are of trypanosomal origin. They probably are but this has not been demonstrated. The statement also implies that these structures have functional roles, which they may, but equally we cannot be certain that they do not represent an aberrant morphological feature, perhaps a stress response, in the population. I think the authors

need to be more equivocal here, the images present are interesting and novel and but at present the significance and role of these structures are uncertain. A clearer statement to this effect should be present in the discussion.

However, we do not want to cast doubt on the parasite origin of the EVs, as these perfectly match observations in *in vitro* axenic cultures (Szempruch, A. J. *et al.*, Cell, 2016; Hempelmann, *et al.*, Cell Rep., 2021). Although we see EVs as beads on a string between proximal parasites, we agree that we cannot formally claim communication. We have slightly modified the text in the discussion to accommodate the reviewer's comment.

A last minor point is line 241

Our present research has identified the lungs as a prominent site of parasite proliferation during natural

The data and the relative parasite load do not justify the term prominent, the authors have shown that the lungs are a site of infection but clearly the parasite load is the same as in the vasculature.

We have removed the term "prominent".